# Statistical parametrization of cell cytoskeleton reveals lung cancer cytoskeletal phenotype with partial EMT signature

Arkaprabha Basu [1], Manash K. Paul[2,3], Mitchel Alioscha-Perez[4,5], Anna Grosberg[6,7], Hichem Sahli[4,5], Steven M. Dubinett[2,3,8,9,10,11] & Shimon Weiss [1,10,12]✉

Epithelial–mesenchymal Transition (EMT) is a multi-step process that involves cytoskeletal rearrangement. Here, developing and using an image quantification tool, Statistical Parametrization of Cell Cytoskeleton (SPOCC), we have identified an intermediate EMT state with a specific cytoskeletal signature. We have been able to partition EMT into two steps: (1) initial formation of transverse arcs and dorsal stress fibers and (2) their subsequent conversion to ventral stress fibers with a concurrent alignment of fibers. Using the Orientational Order Parameter (OOP) as a figure of merit, we have been able to track EMT progression in live cells as well as characterize and quantify their cytoskeletal response to drugs. SPOCC has improved throughput and is non-destructive, making it a viable candidate for studying a broad range of biological processes. Further, owing to the increased stiffness (and by inference invasiveness) of the intermediate EMT phenotype compared to mesenchymal cells, our work can be instrumental in aiding the search for future treatment strategies that combat metastasis by specifically targeting the fiber alignment process.

[1] Department of Chemistry and Biochemistry, University of California Los Angeles, Los Angeles, CA, USA. [2] Department of Medicine, University of California Los Angeles, Los Angles, CA, USA. [3] Division of Pulmonary and Critical Care Medicine, University of California Los Angeles, Los Angeles, CA, USA. [4] Electronics and Informatics Department, Vrije Universiteit Brussel, Brussels, Belgium. [5] Interuniversity Microelectronics Centre, Heverlee, Belgium. [6] Department of Biomedical Engineering, University of California Irvine, Irvine, CA, USA. [7] The Edwards Lifesciences Center for Advanced Cardiovascular Technology, University of California Irvine, Irvine, CA, USA. [8] Department of Pathology and Laboratory Medicine, David Geffen School of Medicine at UCLA, Los Angeles, CA, USA. [9] Department of Molecular and Medical Pharmacology, David Geffen School of Medicine at UCLA, Los Angeles, CA, USA. [10] California NanoSystems Institute, Los Angeles, CA, USA. [11] VA Greater Los Angeles Health Care System, Los Angeles, CA, USA. [12] Department of Physiology, University of California Los Angeles, Los Angeles, CA, USA. ✉email: sweiss@chem.ucla.edu

Metastatic disease is the leading cause of cancer death. Although there are more than 100 different variations of cancer, certain hallmarks are consistent across different malignancies. A healthy tissue can develop a primary tumor based on genetic mutations[1] that can sustain proliferative signaling while resisting growth suppressors and apoptosis. With the formation of a new blood circulation system (angiogenesis)[2], the primary tumor starves the neighboring healthy tissue and due to their uninhibited growth, the cancer tissue achieves replicative immortality. Cells from the primary tumor can eventually metastasize and start secondary tumors[3]. Metastasis often creates insurmountable difficulties in developing treatment strategies and contributes to the high mortality rates.

Numerous studies have enhanced our understanding of metastasis and its markers[4–6], which have demonstrated that metastasis is a complex process involving myriad cellular transformation and migration events. One of the key steps in metastasis is epithelial–mesenchymal transition (EMT), through which the cells in epithelial tissue are transformed into a highly invasive mesenchymal phenotype[7,8]. Epithelial cells are characterized by cell–cell adhesion and apical-to-basal polarity, both of which are lost during EMT[9]. This process involves the downregulation of proteins such as E-cadherin and cytokeratin with a concurrent increase in expression levels of proteins such as N-cadherin, Snail, and Slug. Cells undergoing EMT are known to demonstrate heightened drug resistance properties[10–12]. The EMT program in cancer metastasis co-opts the normal physiological processes in embryonic development[13] and wound healing.

Cytoskeletal rearrangement accompanies EMT, in which cortical actin[14] is re-organized into highly aligned actin stress fibers[15]. This, in turn, plays an important role in changing the elasticity and migration capabilities of the tumor cells. This altered elasticity of mesenchymal cells is essential for movement through constricted space within the tumor microenvironment facilitating access to the vasculature. The corresponding cytoskeletal rearrangement is conserved across most solid malignancies. A more complete understanding of the interrelation between the EMT genetic program and stress fiber formation is required.

The cytoskeletal reorganization requisite for the formation and alignment of stress fibers during EMT will advance our understanding of metastatic behavior. Stress fibers are responsible for maintaining cell shape, aiding in cell migration and intracellular cargo transport. These fibers are a complex moiety consisting of actin filaments held together by actin-binding proteins (ABPs) such as myosin, α-actinin, and filamin[16,17]. Based on the presence and localization of specific ABPs, the stress fibers can have vastly different structures and functions, such as transverse arcs, dorsal fibers, and ventral fibers[18–22]. Monitoring the formation of stress fibers in EMT and correlating them with the corresponding ABPs could be utilized to develop a new and reliable reporter for EMT and thus develop screens for inhibitors of the EMT process.

There have been multiple approaches to study and track cellular EMT, employing simple biochemical experiments and mass cytometry studies[23,24] to analyze the regulation of EMT marker proteins as well as single-cell RNA sequencing techniques[25]. These studies have confirmed the existence of intermediate states with partial EMT phenotypes, that are neither entirely epithelial or mesenchymal, based on marker proteins and gene expression levels[23]. However, such techniques can be expensive, low throughput, and involve cellular destruction, which prevents temporal assessments. Here, we propose an imaging-based non-destructive method for quantifying the cytoskeletal changes accompanying EMT in live cells which provides a unique tool with potentially improved throughput for tracking EMT progression in real time.

We are focusing on lung cancer metastasis, the leading cause of cancer deaths worldwide[26,27]. In this study, we propose that the epithelial cells with cortical actin (C) traverse one or more intermediate states (I) before reaching a final mesenchymal state with aligned stress fibers (A). Because the final aligned state is reported to have enhanced invasiveness, it would be advantageous to intervene at the intermediate states in order to prevent invasion. We further propose that the formation of stress fibers is a separate event from fiber alignment and as such the intermediate states should have nonaligned stress fibers. We have exploited the sequential formation of different types of stress fibers as a key for identifying a non-binary EMT state that exists as an intermediate between the epithelial and mesenchymal phenotypes. This intermediate phenotype is characterized by a disorganized actin cytoskeleton and predominantly different stress fibers compared to normal mesenchymal cells. Actin stress fibers can be treated as a composition of quasi-straight elements, which will have a distinctive geometric pattern from other artifact structures and noise. We have developed a tool called Statistical Parametrization of Cell Cytoskeleton (SPOCC) where we have used previously described tools[28–30] to extract the geometry of the actin cytoskeleton as a series of straight lines with their corresponding locations, lengths, and angles. We used the angular distribution of the cell cytoskeleton to calculate the Orientational Order Parameter (OOP)[31–33], which serves as a figure of merit for stress fiber alignment (enabling us to identify the C, I, and A states) as well as EMT progression. We confirmed the viability of OOP as an EMT reporter by inhibiting EMT using multiple drugs to arrest the alignment of fibers. We have also correlated the intermediate phenotype with lower stiffness compared to mesenchymal cells (stiffer than epithelial cells) supporting their partial EMT properties.

## Results

**Cells in the early phases of EMT demonstrate a previously unreported cytoskeletal architecture distinct from later phases.** Because the formation of actin stress fibers is a well-established phenomenon as cells undergo EMT (Fig. 1a), we sought to understand the evolution of the cellular cytoskeleton during EMT. There is a growing recognition that EMT is a non-binary process, and there have been previous studies identifying the intermediate states involved using mass cytometry and single-cell RNA sequencing[23]. Due to the extensive interconnectedness between the genetic pathways responsible for EMT and actin stress fiber formation, the intermediate partial EMT states are likely to have their own cytoskeletal signatures. We chose A549, H460, and H1299 cell lines because they are well-established lung cancer models. To study the sequential evolution of stress fibers, we induced EMT in the cells using Transforming Growth Factor-β1 (TGFβ1) and fixed them at specific time intervals up to 48 h after the addition of TGFβ1 (Supplementary Fig. 1a). Initially, there was no stress fiber in most cells (Fig. 1b). We observed that cells at the later timepoints had well-aligned (semi-parallel) stress fibers consistent with the mesenchymal phenotype (Fig. 1d). In contrast, at earlier timepoints, stress fibers were observed, but they were completely disorganized (Fig. 1c). The progress of EMT was also confirmed by tracking the expression levels of E-cadherin and N-cadherin with time in A549 cell. Not only was E-cadherin almost completely lost in the cells treated with TGFβ1 for 48 h confirming their mesenchymal nature, after 14 h of TGFβ1 treatment, there was still a significant amount of E-cadherin (though it is less than untreated cells), indicating at a partial EMT nature of the earlier cytoskeletal phenotype. Expression of Vimentin, Slug, and N-cadherin was upregulated as a function of time with the earlier time-points having intermediate levels of expression (Supplementary Figs. 1b, c and 2).

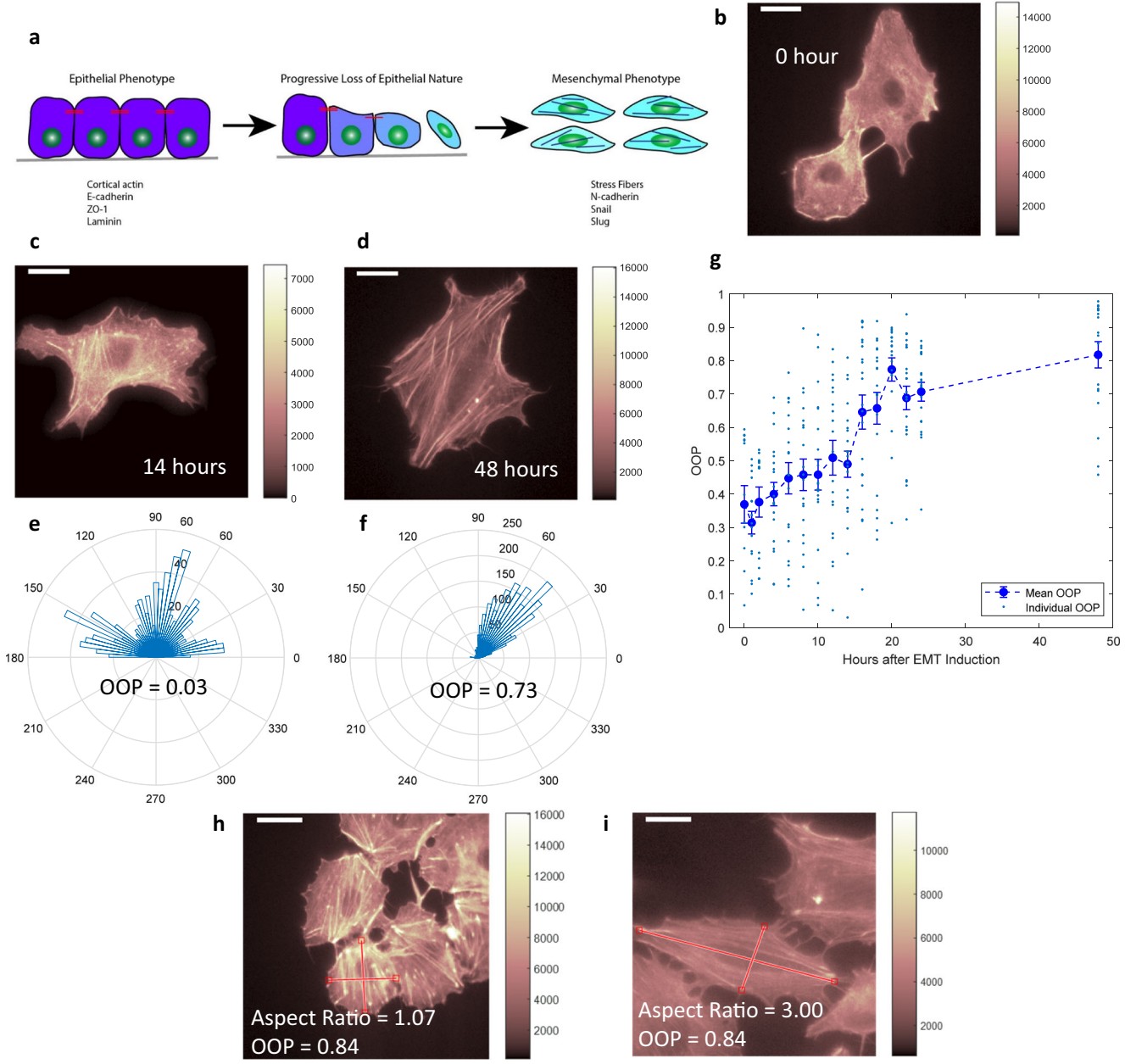

**Fig. 1 Identification and quantification of cytoskeletal phenotypes. a** Cartoon image of cells undergoing EMT with formation of stress fibers and up/downregulation of proteins. **b-d** Fluorescent images of A549 cells stained with phalloidin after 0, 14, and 48 h of TGFβ1 induced EMT, respectively. **e**, **f** Angular distribution of stress fibers and corresponding Orientational Order Parameter (OOP) values for cells shown in (**c**, **d**), respectively. **g** Plot of OOP values for a cell population against time of TGFβ1 treatment. Mean values are reported and error bars correspond to standard error values for every timepoint. $n = 12$ (for 0 h), $n = 15$ (for 1 h), $n = 13$ (for 2 h), $n = 20$ (for 4 h), $n = 19$ (for 6 h), $n = 20$ (for 8 h), $n = 22$ (for 10 h), $n = 20$ (for 12 h), $n = 22$ (for 14 h), $n = 22$ (for 16 h), $n = 20$ (for 18 h), $n = 19$ (for 20 h), $n = 23$ (for 22 h), $n = 20$ (for 24 h), and $n = 19$ (for 48 h). **h**, **i** Fluorescent images of two cells stained with phalloidin with highly aligned stress fibers (similar OOP values) but drastically difference aspect ratios. Scale bar: 16 μm.

H460 and H1299 cells also demonstrated similar features for early and late phase EMT (Supplementary Fig. 3a–g). Though the stress fibers in individual cells were aligned in the mesenchymal phenotype, different cells in the same region demonstrated different directions of fiber orientation (Supplementary Fig. 3c). To quantify the difference in the angular distribution of stress fibers we developed a technique called Statistical Parametrization of Cell Cytoskeleton (SPOCC), where we extracted the actin filaments as a series of straight lines with their corresponding locations, angles, and their lengths using a morphological component analysis and line segmentation algorithm. We then calculated the Orientational Order Parameter (OOP) from the

angular distributions (Supplementary Fig. 4). A narrow angular distribution of well-aligned stress fibers corresponds to a high OOP value (Fig. 1e) and a broad distribution in disorganized fibers results in a low OOP value (Fig. 1f). The cell population demonstrated alignment of fibers with time in EMT and the OOP value concurrently increased (Fig. 1g), making the OOP value a good phenotypic marker (or "figure of merit") for the alignment of stress fibers as well as the progression of EMT. Previous studies have utilized cell aspect ratios as well as the actin fluorescence intensity as markers for cytoskeletal remodeling[34,35]. But a comparison of simple fluorescence intensity cannot uncover reorganization of the existing cytoskeleton effectively where the

total amount of actin is not changing (Supplementary Fig. 5). Also, the extent of bleaching can vary from cell to cell, making the fluorescence intensity data subject to errors. In the case of the aspect ratio comparison, the major and minor axes of the cells are not always well defined due to their irregular shapes and so the calculation of aspect ratio faces some inherent challenges. Also, cells with well-aligned fibers (high OOP cells) can have completely different aspect ratios (Fig. 1h, i and Supplementary Fig. 6). Thus, the OOP value calculated using SPOCC is a more relevant cell state marker during the EMT and can extract more information from similar fluorescent images than existing methods.

**Early- and late-stage EMT cells have predominantly different types of stress fibers.** It can be inferred from the alignment of stress fibers that EMT is a continuous process where first the stress fibers are formed throughout the cell in different orientations and subsequently align to produce the final phenotype. In this case, the cells with nest-like architecture would be a single point-in-time snapshot of an undefined point along the transition pathway. In order to verify that the two architectures were distinct phenotypes, we investigated the nature of their stress fibers. Based on the presence and localization of actin-binding proteins (ABPs), the stress fibers can have completely different morphology and function (Fig. 2a). Focal Adhesion Kinase demonstrates the most distinct localization patterns across different stress fiber types[18,19]. We stained cells for both actin (Fig. 2b, e and Supplementary Figs. 7 and 8) and FAK (Fig. 2c, f) and compared the patterns in the two phenotypes. Cells in early EMT had FAK spots predominantly around the cell edge compared to cells in late EMT which had FAK spots throughout the cell. From the overlay images (Fig. 2d, g) we observed that the stress fibers in the late EMT stage cells were capped on both ends with FAK. In contrast, the early EMT cells had stress fibers with only one or neither of their ends FAK-capped.

**Identifying and quantifying a phenotypic transition in single-cell EMT trajectories.** Two possible models may explain the existence of the two phenotypes in early and later stage EMT. In the first model, the two phenotypes represent two separate cell populations resulting from different parts of the EMT genetic cascade that were activated at different timepoints. In such a case, the low OOP cells will retain their low OOP values throughout EMT and at later timepoints, high OOP cells will start appearing in the population; so though there will be an overall increase in OOP of the population, OOPs of individual cells will not change significantly with time. A second model describes the disorganized phenotype transitioning into a higher degree of alignment with the progression of EMT, resulting in an increase in OOP of individual cells. To determine which of the two proposed models is operative, we tracked single cells (stained with SiR-Actin) (Fig. 3a, b) undergoing EMT over time. The gradual increase in the OOP value (Fig. 3g, Supplementary Fig. 9, and Supplementary Video. 1) suggests a phenotypic transition rather than two independent and distinct populations.

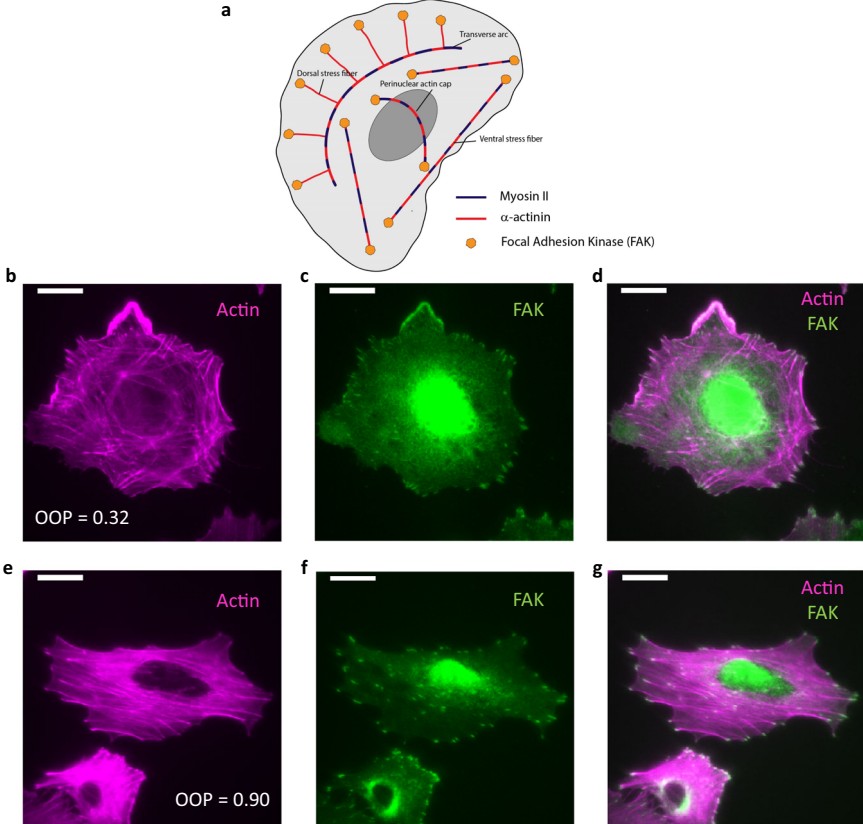

**Fig. 2 Focal Adhesion Kinase (FAK) pattern reveals types of stress fibers. a** Cartoon representation of stress fibers and actin-binding proteins. **b** Fluorescent image of a cell with disoriented actin stress fibers (OOP = 0.32). **c** Fluorescent image of FAK of the same cell shown in (**b**) showing FAK spots near the cell edge. **d** Overlay image of actin (magenta) and FAK (green) of the same cell shown in (**b**) showing stress fibers with zero or one FAK capping. **e** Fluorescent image of cell with semi-parallel actin stress fibers (OOP = 0.90). **f** Fluorescent image of FAK of the same cells shown in (**e**) showing FAK spots throughout the cell. **g** Overlay of actin (magenta) and FAK (green) for the same cell shown in (**e**) showing FAK spots on both ends of stress fibers. Scale bar: 16 μm.

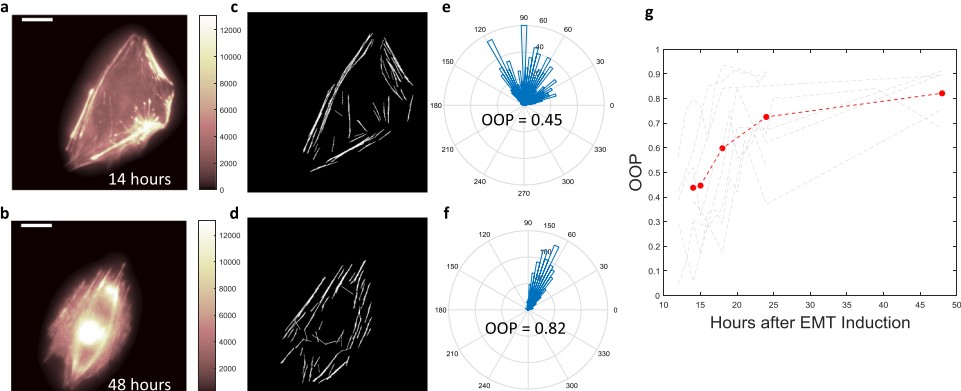

**Fig. 3 Tracking phenotypic transition in single cell stained with SiR-actin. a, b** A single A549 cell stained with SiR-actin after 14 h and 48 h of TGFβ1 addition respectively. **c, d** Extracted stress fiber image from (**a, b**), respectively. **e, f** Angular distribution and OOP values of the cells shown in (**a, b**), respectively. **g** Plot of OOP values of a live cell (shown in (**a–f**)) against time of TGFβ1 treatment (red). Multiple live-cell OOP trajectories (gray) against the time of TGFβ1 treatment. Scale bar: 16 μm.

**Phenotypic transition responds to EMT pathway inhibition.** Multiple genetic pathways are involved in EMT. These pathways can operate consecutively or in parallel and each of these pathways have different levels of cross-talk with the formation of stress fibers. As the stress fiber alignment is a distinct process from the formation of stress fibers, we can expect each step to be controlled by a different part of the EMT signaling cascade. We sought to verify the two-step nature of EMT by differentially affecting the two steps using known pathway inhibitors for EMT. First, we inhibited the Rho-ROCK pathway, which is one of the most well-known EMT pathways[36–41]. When we inhibited this pathway with Rhosin[42], it resulted in a complete suspension of stress fiber formation in the drug-treated cells after 48 h of TGFβ1 and inhibitor treatment (Fig. 4b). To quantify the suspension of stress fiber formation, we compared the number of extracted fibers as well as the total length of fiber extracted in untreated cells vs inhibitor-treated cells (Fig. 4e and Supplementary Fig. 10). We observed that inhibitor-treated cells had demonstrably fewer extracted fibers. The Wnt pathway[43–46] was also assessed by inhibition utilizing two different methods. We used XAV 939 to inhibit Tankyrase1/2[47,48] and JNK-IN-8 to inhibit c-Jun N-terminal kinase 1/2 (JNK 1/2)[49] both of which are involved in the Wnt pathway. With both these inhibitors, the cells demonstrated a disorganized stress fiber arrangement after 48 h of TGFβ1 and inhibitor treatment (Fig. 4c, d, f). We then evaluated the ability and accuracy of SPOCC in characterizing the drug response. We calculated the OOP values for a series of cell populations undergoing EMT with increasing time of TGFβ1 and XAV 939 treatment and compared them with OOP values of cell populations without the presence of XAV (Fig. 4g and Supplementary Fig. 11). We found that the OOP values for cell populations treated with XAV 939 did not show the same increase with time that the untreated cell populations showed. We also followed single cells undergoing EMT with and without the presence of XAV 939 and calculated their OOP values at multiple timepoints (Fig. 4h). We established that even at the single-cell level the OOP values do not increase with time upon inhibitor treatment. These findings corroborate our hypothesis that inhibitor treatments can selectively arrest the phenotypic transition.

**The early EMT phenotype demonstrates a partial EMT nature and has different elastic properties compared to mesenchymal cells.** Lung cancer mesenchymal cells are known to be stiffer compared to epithelial cells[50]. Due to the extensive interrelationship between the actin cytoskeleton and elastic properties of cells, the nest-like phenotype can also be expected to demonstrate a partial epithelial nature and be more compliant compared to the mesenchymal phenotype. To assess this relationship, we conducted atomic force microscopy, performing force curve measurement experiments to calculate the elastic modulus (Young's modulus) of the three phenotypes. The epithelial cells had the lowest Young's modulus and the late EMT mesenchymal phenotype had the highest Young's modulus. The Young's modulus of the nest-like phenotype was intermediate between the two (Fig. 5).

## Discussion

To better understand the cytoskeletal rearrangements involved in EMT we tracked EMT progression in lung cancer cells. We identified a phenotype, which was previously unreported to the best of our knowledge, in which the stress fibers were not aligned in any particular direction. This alignment process was also identified in single cells by tracking them through EMT. Thus, we identified the alignment process as a phenotypic transition. These results indicate that EMT is at least a two-step process, the first step being the formation of stress fibers followed by their alignment (Fig. 6). The nest-like phenotype is an intermediate along the pathway. The lack of a common direction of alignment of stress fibers from cell to cell is indicative that the alignment process is likely not influenced by the availability of space in a particular direction. Cells respond to matrix stiffness by altering their own mechanical properties[51,52]. This disoriented stress fiber architecture with no dominant direction of alignment was previously reported for cells grown on soft surfaces[34]. The other feature that accompanied the orientation of stress fibers was their types. The difference in their fiber alignment is indicative of a difference between the stiffness of the two phenotypes. Our AFM experiments corroborate the hypothesis that difference in cytoskeletal architecture results in altered mechanical properties. We demonstrated that the nest-like phenotype occupies an intermediate elastic niche between epithelial and mesenchymal cells. Earlier studies have reported a decrease in cell stiffness as cells undergo EMT[53–56]. But recently, it has been demonstrated that in the case of pre-invasive breast cancer and non-small cell lung cancer there is a concurrent increase in cell stiffness/rigidity with EMT progression resulting from regulation of motor proteins[50,57]. A possible explanation is that EMT induced by growth factors have different physiological effects resulting in cell stiffening. Growth factor concentration is usually highest at the tumor margins from where the cells begin to migrate, therefore growth factor exposure in vitro may reflect the in vivo

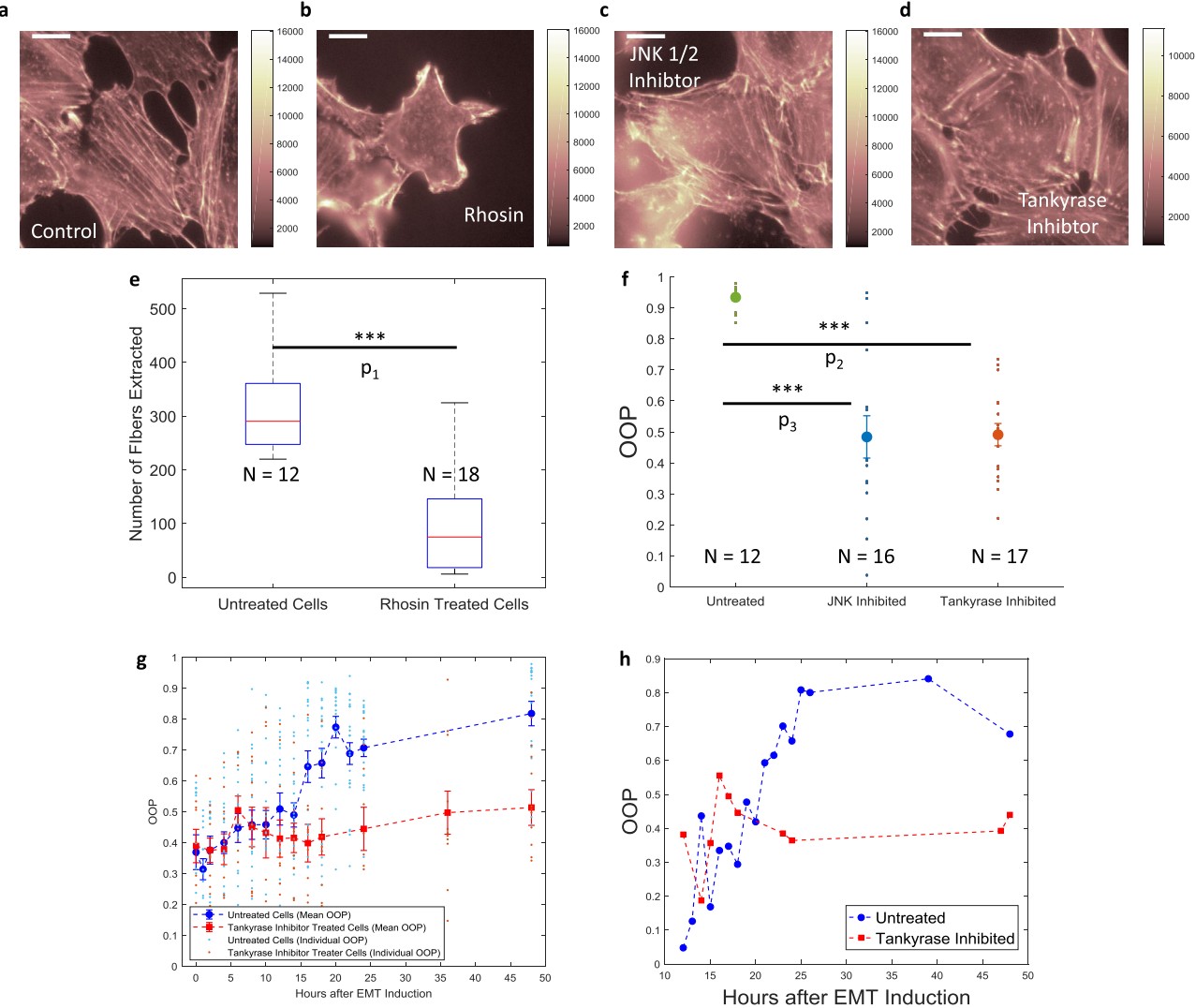

**Fig. 4 Quantification of drug response of EMT over 48 h. a–d** A549 cells stained with phalloidin after 48 h of EMT induction in the presence of no drug (**a**), Rhosin (**b**), JNK 1/2 Inhibitor (**c**), and Tankyrase Inhibitor (**d**). **e** Boxplot comparison of the number of stress fibers extracted from control (no drug) cells vs Rhosin treated cells. Red lines inside the boxes correspond to the median values, the bottom and top edges of the boxes correspond to the 25th and 75th percentiles respectively, the whiskers are extended to the most extreme data that is not considered an outlier in each direction. **f** Plot showing the OOP values of control cells (green) and cells treated with JNK 1/2 inhibitor (blue) and Tankyrase inhibitor (red). Mean of the distributions is shown as larger dots of the same color and the error bars correspond to standard error values. **g** Plots of individual and mean OOP values against the time of a cell population undergoing EMT with (red) and without (blue) the presence of Tankyrase Inhibitor. Error bars correspond to standard errors at each timepoint. For untreated cells, the sample size at each timepoint is same as reported in Fig. 1g. For Tankyrase inhibited cells: $n = 9$ (for 0 h), $n = 12$ (for 2 h), $n = 11$ (for 4 h), $n = 9$ (for 6 h), $n = 11$ (for 8 h), $n = 11$ (for 10 h), $n = 11$ (for 12 h), $n = 11$ (for 14 h), $n = 10$ (for 16 h), $n = 12$ (for 18 h), $n = 10$ (for 24 h), $n = 11$ (for 36 h), and $n = 9$ (for 48 h). **h** Plots of OOP values against the time of a single cell undergoing EMT with (red) and without (blue) the presence of Tankyrase Inhibitor. Scale bar: 16 μm. *$P < 0.05$, **$P < 0.01$, ***$P < 0.001$. $P_1 = 1.85 \times 10^{-6}$, $P_2 = 1.28 \times 10^{-10}$, $P_3 = 6.24 \times 10^{-6}$.

environment at the tumor margins. It is also possible that this phenomenon is unique to certain types of cancer cells. Cells in the lung and airways are exposed to constant expansion and compression which thus requires a compliant lung epithelium. Upon EMT induction, the remodeling of the cytoskeleton promotes elevated rigidity. Our findings suggest that the stiffness of the intermediate phenotype is a property indicative of its partial epithelial nature which is intermediate between the epithelial and mesenchymal phenotypes. As the cytoskeleton extensively affects the elasticity and motility of cells, we can expect the motility and invasiveness of the intermediate phenotype to be in between that of the epithelial and mesenchymal phenotypes.

Though one of the hallmarks of EMT is the loss of cell–cell adhesion through downregulation of E-cadherin, our data

demonstrate that a few cell clusters exist even after EMT. This means cells can undergo EMT without complete loss of cell–cell adhesion indicating that the downregulation of E-cadherin can be nonuniform and the extent of the downregulation can vary between different cell types. Reports of collective migration of tumor cells[58] also support the theory that EMT can take place without complete loss of cell–cell adhesion. Future studies can investigate how E-cadherin downregulation is regulated across different cell types and what role it plays in EMT.

To quantify this cytoskeletal phenomenon, we developed an image analysis and quantification technique, Statistical Parametrization of Cell Cytoskeleton (SPOCC), that can identify and differentiate between the phenotypes from simple fluorescent images of the cell cytoskeleton. Though recently reported

techniques can extract similar information about the stress fibers[59,60], our technique assigns a figure of merit for the relative alignment of fibers to individual cells. OOP has been used traditionally to quantify the alignment of cells in tissue environment[31], but here we have proposed using OOP as a measure of relative alignment of the cytoskeleton. In most works where the OOP has been used to analyze the cytoskeleton[32,61–63], the individual pixel orientations are calculated using FFT (fast Fourier transform) or pixel intensity gradient[64] and the OOP is calculated from pixel-based orientation vectors. In certain cases, the lengths of fibers are calculated based on position of actin-binding proteins. The actin extraction algorithm used in SPOCC is more robust than FFT and can extract information on the length, position, and orientation of individual fibers as well as OOP based on just actin images. As SPOCC is capable of extracting and quantifying multiple properties of the cytoskeleton, it is better suited for biological processes such as EMT where the combination and correlation of multiple aspects of the cytoskeleton can uncover more information. Beyond EMT, SPOCC is also capable of quantifying other biological processes that involve cytoskeletal remodeling and may have broad applications.

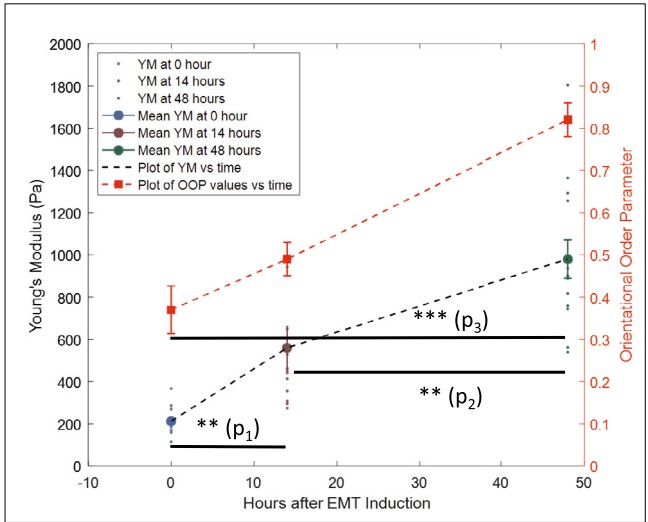

While migration and motility assays directly measure biophysical properties (such as invasiveness) of cells, these measurements average the properties over a long period of time making them incapable of identifying or tracking faster biological processes. SPOCC, on the other hand, is limited only by microscopic imaging and as such can provide much better time resolution, but cannot directly measure the biophysical properties of the cells. Given the comprehensive interdependence of cytoskeletal structure and motility, SPOCC provides the perfect means of generating a library correlating the cytoskeletal structure of cells with their motility (and possibly other biophysical properties). Such a library would enable future studies to estimate the motility of cells with better time resolution based on the SPOCC data.

Transverse arcs do not have any FAK capping whereas dorsal and ventral stress fibers have one and both ends FAK-capped respectively[18,19]. Based on the FAK patterns in the two phenotypes, we identified the stress fiber types in each phenotype. We demonstrated that the mesenchymal phenotype has ventral stress fibers whereas the intermediate phenotype predominantly has dorsal fibers and transverse arcs. It has been reported in the literature that two dorsal stress fibers or a combination of dorsal stress fibers and transverse arcs can form ventral stress fibers[16,65,66]. Based on the dependence of the ventral fiber formation and cellular migration, we believe a very similar mechanism is operative in the case of EMT, which is known to increase the motility of cells. The enhancement of ventral stress fibers results in better anchoring on the substrate. This in turn is likely to increase the motility of the cells. Also, as the stress fiber mesh moves to the ventral side of the cells with progression of EMT, the elastic properties of the cells are likely to change as well.

As anticipated, our results demonstrate that either the first step or both the steps, as discussed above, are dependent on the Rho-GTPase (Rho-ROCK) pathway. The Wnt pathway is involved in the stress fiber alignment process but not their formation. Along with its reported role in stress fiber alignment[63], JNK is also involved in the p38-MAPK pathway[67,68], which is known to have comprehensive cross-talk with the Wnt pathway[46]. Inhibition of the Wnt pathway alone with tankyrase resulted in similar outcomes which suggest the involvement of the Wnt/p38-MAPK pathway in the fiber alignment process. We anticipate that the stress fiber alignment is carried out in conjunction with a kinase controlled by the Wnt/p38-MAPK pathway. The identification and silencing of this kinase may be a valuable tool for controlling similar biological processes. Further studies will be required to definitively define the requisite pathways.

Previous studies have reported JNK/ERK-mediated stress fiber alignment in cells undergoing cyclic stretching[63]. Though there is no active stretching of the cell (or the substrate) in EMT, it is important to understand how cells might perceive stretching. The stretching process induces different levels of tension in the cell along the direction of stretching and the perpendicular direction. Epithelial cells show apical-to-basal polarity which is lost during

**Fig. 5 Measurement of elastic properties of cells.** Young's Modulus values at 0 (blue) ($n = 15$), 14 (brown) ($n = 13$), and 48 (green) ($n = 14$) hours after EMT induction using TGFβ1 showing their mean and standard errors along with a plot of the mean values. Young's modulus values were extracted from AFM force curve measurements. Plot of mean OOP values for cell populations against hours after EMT (red) showing mean values and standard errors at 0, 14, and 48 h timepoints. Sample sizes for OOP values are same as reported in Fig. 1g. $*P < 0.05$, $**P < 0.01$, $***P < 0.001$. $P_1 = 0.0012$, $P_2 = 0.0048$, $P_3 = 3.6 \times 10^{-9}$.

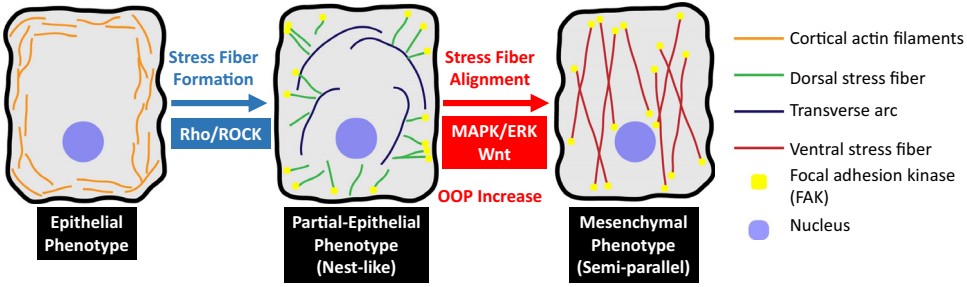

**Fig. 6 Schematic of cytoskeletal reorganization in EMT.** Schematic of the cytoskeletal reorganization, relevant genetic pathways and increase of OOP during Epithelial–Mesenchymal Transition of A549 cells.

**Table 1 Primary and secondary antibodies used along with their information.**

| Antibody name | Company | Catalogue no. | Primary dilution | Incubation time | Incubation temperature (°C) |
|---|---|---|---|---|---|
| GAPDH | Advanced Immuno Chemical Inc | 2-RGM2 | 1:10,000 | Overnight | 4 |
| β-actin | Cell Signaling Technology | 12262 | 1:10,000 | Overnight | 4 |
| E-cadherin | BD Transduction Laboratories | 610182 | 1: 400 | Overnight | 4 |
| N-cadherin | BD Transduction Laboratories | 610920 | 1:400 | Overnight | 4 |
| Slug | Cell Signaling Technology | 9585 | 1: 300 | Overnight | 4 |
| Goat anti-mouse IgG | LI-COR Biosciences | 926-32210 | 1:5000 | 60 min | Room temperature |
| Goat anti-rabbit IgG | LI-COR Biosciences | 926-68071 | 1:5000 | 60 min | Room temperature |

EMT. As the cells start losing their polarity, the change in the tension that the cell experiences in different along axis of polarity and its perpendicular direction, essentially mimicking the stretching condition. We hypothesize that this loss of polarity (coupled with migration) can induce a similar effect in the cell as external stretching and is responsible for fiber alignment.

As stress fibers are involved in multiple functions in healthy cells, it is unlikely that the complete termination of stress fiber formation is a viable clinical approach to counter metastasis. However, arresting the second step, the alignment process, alone may allow for the proper functioning of normal cells. Inhibiting the alignment process may thus impede cell migration and prevent metastasis.

To summarize, in this work, we discovered a partial EMT phenotype in lung cancer cells with a unique cytoskeletal signature which are consistent with decreased stiffness compared to mesenchymal cells. We have partitioned the cytoskeletal component of EMT into two separate steps: (1) the formation of stress fibers and (2) the alignment of stress fibers. We have also demonstrated that it is possible to arrest the alignment process selectively by inhibiting the Wnt pathway. We have developed SPOCC, an image quantification technique that can identify and differentiate between different cytoskeletal morphologies from simple fluorescent images.

In future studies, we will evaluate EMT in a broader spectrum of cell lines to further our understanding of partial epithelial phenotypes in the context of different lung cancer driver mutations. Correlating the time evolution of the transcriptome with the increase in OOP, followed by subsequent silencing of key genes, may define additional mechanisms operative in this process.

In conclusion, we have demonstrated that accurate assessments of cytoskeletal dynamics can inform our understanding of the determinants of EMT progression providing biological data potentially relevant in future clinical applications.

## Methods

**Cell culture**. All cell lines (A549, H460, and H1299) were purchased from ATCC. A549 cells were cultured in DMEM (Gibco, catalog no. 11995-065) supplemented with 10% FBS (Gibco, catalog no. A31604-01) and 1% penicillin streptomycin (10,000 U/ml, Gibco, catalog no. 15140-122). H460 and H1299 cells were cultured in RPMI (Gibco, catalog no. A10491-01) supplemented with 10%FBS (Gibco, catalog no. A31604-01) and 1% penicillin streptomycin (10,000 U/ml, Gibco, catalog no.15140-122). EMT in all cells lines was induced by the addition of 5 ng/ml Targeted Growth Factor-β1 (TGFβ1) (Peprotech, catalog no. 100-21-10UG) for 48 h[69].

**Western blot analysis**. Total cell lysates were prepared and western blots were done as reported earlier[70] using the primary and secondary antibodies (Table 1). BCA method was used for the estimation of protein concentrations using the manufacturer's guidance. An equal volume of 2× SDS sample buffer was added and the samples were denatured by boiling for 5 min. Samples were applied to an SDS-PAGE and transferred to an Immobilon PVDF membrane (Millipore, USA). The membranes were blocked with 5% skimmed milk prepared using Tris-buffered saline with 0.05% Tween 20, and then treated with primary antibodies. The membranes were incubated with primary antibodies overnight at 4 °C. The membranes were then rinsed three times with Tris-buffered saline containing 0.1%

**Table 2 EMT inhibitors used along with their information.**

| Target protein | Drug name | Catalog no. (Selleckchem.com) | Concentration |
|---|---|---|---|
| Rho-GTPase | Rhosin hydrochloride | S8988 | 0.4 μM |
| JNK 1/2 | JNK-IN-8 | S4901 | 20 nM |
| Tankyrase1/2 | XAV-939 | S1180 | 11 nM |

Tween 20 (TBST) after incubation with primary antibodies. The membranes were then incubated in TBST containing 5% BSA for 1 h with horseradish peroxidase-conjugated goat anti-mouse IgG and horseradish peroxidase-conjugated goat anti-rabbit IgG secondary antibodies (LI-COR Biosciences, Lincoln, NE). After that, the blots were washed three times in TBST and the immune complexes were visualized with the ECL kit (GEHealthcare, USA)[71]. Proteins were observed and scanned using an Odyssey Infrared Imaging System (LI-COR Biosciences, Lincoln, NE) with 700- and 800-nm channels to scan the membrane. As internal loading controls, the blots were re-probed with anti-GAPDH or anti-actin antibodies. ImageJ software was used to compute the relative densitometry values. Band intensity was also quantified by ImageJ software (Rasband,1997–2014). The obtained images were converted to 8-bit format and then subjected to background subtraction through the rolling ball radius method. Quantification of peak area of obtained histograms was performed for each individually selected band. All western blots were performed independently in triplicates and the data are represented as standard error of the mean (SEM) for all performed repetitions[72]. Internal loading controls were used to normalize the data. The results from the untreated groups were used to calculate relative values.

**Cell fixing (endpoint study)**. Cells were grown on eight-well culture slides (Sarstedt, Catalog no. 94.6170.802) for 24 h and treated with 5 ng/ml TGFβ1 (and drugs) for 48 h. After 48 h, cells were rinsed with PBS (Gibco, catalog no. 14190-136) and fixed with 4% paraformaldehyde (diluted from 10%) (Electron Microscopy Sciences, catalog no. 15712-S) for 20 min.

**Time-point study**. Cells were grown on fibronectin-coated cover slides (neuVitro, catalog no. GG-12-fibronectin) for 24 h and treated with 5 ng/ml TGFβ1 (and drugs). At specific time intervals after TGFβ1 treatment, the cover slides were rinsed with PBS (Gibco, catalog no. 14190-136) and fixed with 4% paraformaldehyde (diluted from 10%, Electron Microscopy Sciences, catalog no. 15712-S) for 20 min.

**Drug treatment**. Cells were grown for 24 h before being treated with 5 ng/ml TGFβ1 and specific drugs (Table 2) for specified times (48 h for endpoint experiments).

**Cell staining**. Fixed cells permeabilized with 0.1% Triton X-100 (Research Products International Corp., catalog no. 11036) for 5 min, blocked with freshly prepared 5% BSA (Fisher BioReagents, catalog no. BP 1600-100, CAS no. 9048-46-8) for 25 minutes, treated with 1:100 solution of primary antibody in blocking medium at 4 °C overnight. Next, the cells were rinsed thoroughly and stained with 1:200 solution of secondary antibody in PBS for 2 h followed by staining with Acti-Stain™ 670 Fluorescent Phalloidin (Cytoskeleton Inc., catalog no. PHDN1). Then the cover slides were mounted on glass slides using ProLong Diamond Antifade Mountant (Invitrogen, catalog no. P36961) and sealed with clear nail polish. Primary antibodies: Anti-FAK (D1) mouse monoclonal IgG$_1$ antibody (Santa Cruz Biotechnology, catalog no. sc-271126). Secondary antibody: Alexa Fluor® 488 AffiniPure F(ab')$_2$ Fragment Donkey Anti-Mouse IgG (H + L) (Jackson ImmunoResearch Laboratories Inc., code. 715-546-151).

**Live-cell staining**. A549 cells were grown on glass-bottomed dishes (Cellvis, catalog no. D35-20-1.5-N) for 24 h and treated with 5 ng/ml TGFβ1 and 100 nM SiR-actin kit (Cytoskeleton Inc., CY-SC001).

**Fluorescence imaging**. Fluorescent cells were imaged on a Nikon Ti-Eclipse microscope equipped with an AURA light engine (Lumencor) light source and ×60 oil immersion objective lens (Nikon, Plan Apo VC 60X/1.4). The images were captured using a iXon+ camera (Andor Technologies, model no. DU-897E-CSO-#BV). For live-cell imaging, a microscope mounted incubator (Warner Instruments Inc., model no. DH-40iL) coupled with an automatic temperature controller (Warner Instruments, Inc., model no. TC-324C) was used, which kept the cells at 37 °C, 5% $CO_2$, and 90% relative humidity.

**Image analysis and fiber extraction**. Fluorescent images were processed using Matlab (version 2016a) and analyzed using previously described protocol[30]. In this algorithm, the fluorescence image is treated as a sum of three components: (1) the filaments image, (2) the artefacts image, and (3) noise, where the primary goal is to separate the filaments image from the artefacts and noise. This is achieved by exploiting the fact that the filaments have a quasi-straight morphology that is unlikely to be randomly created as artefact or noise. So the filaments are extracted through a curvelet transform, whereas the artefacts were extracted using an undecimated wavelet transform provided by the MCALab libraries[73] and running 100 iterations. The filament image was enhanced to improve the contrast and sharpen the edges by using a sequence of filters: (1) a Gaussian filter, (2) a Laplace filter, and (3) a directed Gaussian filter. Next, a multi-scale line segmentation step assigns a probability to every pixel of being part of a line of a certain width by evaluating its neighborhood. Wellner's adaptive thresholding is used to binarize the image. For extracting individual straight lines (filaments), a line segmentation step is used on the binary image to fit a straight line of a given minimum length (L = 30 for the experiments in this paper) to sequential non-zero pixels. Then, overlapping straight lines of the same orientation are stitched together to form longer filaments. This process generates a binary image consisting of straight lines (filaments) whose length, orientation, and location are known.

**Calculating Orientational Order Parameter (OOP)**. The filament angles were extracted from the output and Orientational Order Parameter (OOP) was calculated from the angular distribution[31,32]. OOP is defined as the maximum eigenvalue of the Mean Order Tensor of a set of vectors. First, every angle (vector) is converted into their corresponding tensors.

$$Vector\,(Angle) \xrightarrow{yields} \begin{bmatrix} p_{i,x} \\ p_{i,y} \end{bmatrix} \qquad (1)$$

$$Order\,Tensor = \begin{bmatrix} p_{i,x}p_{i,x} & p_{i,x}p_{i,y} \\ p_{i,x}p_{i,y} & p_{i,y}p_{i,y} \end{bmatrix} \qquad (2)$$

The mean order tensor is calculated from the individual tensors.

$$Mean\,order\,tensor = T = \left\langle 2\begin{bmatrix} p_{i,x}p_{i,x} & p_{i,x}p_{i,y} \\ p_{i,x}p_{i,y} & p_{i,y}p_{i,y} \end{bmatrix} - \begin{bmatrix} 1 & 0 \\ 0 & 1 \end{bmatrix} \right\rangle \qquad (3)$$

The possible eigenvalues and eigenvectors of the mean order tensor are calculated. OOP is the maximum eigenvalue of the mean order tensor.

$$OOP = \max\left[eigenvalue(T)\right] \qquad (4)$$

The OOP calculated here does not take into account the length of fibers, but based on our observations, the lengths of fibers do not vary drastically between cells (Supplementary Fig. 12). As our image analysis software extracts the stress fibers as straight lines, they are chopped up into smaller lengths.

**Atomic force microscopy**. Cells were grown on glass-bottomed dishes (Fluor-oDish, World Precision Instruments Inc.) for 24 h before the addition of 5 ng/ml TGFβ1. AFM force curves were obtained using a Bruker Nanowizard 4A instrument coupled with a Zeiss Observer.Z1 Microscope with LSM5 Exciter laser scanning confocal module and ×40 oil immersion objective lens (Zeiss, EC Plan-NeoFluar 40X/1.3). A nitride tip (Bruker, SAA_SPH-5UM) with a nitride lever was used. All the force curves were analyzed using the JPKSPM Data Processing Software.

**Statistics and reproducibility**. We have carried out the two-sample $t$ tests using the "ttest2" function in the Statistics and Machine Learning Toolbox in Matlab. It returns a test decision regarding the validity of the null hypothesis that the two sets of data come from the normal distribution of equal means and equal variances. The rejection of the null hypothesis is done at 5% significance level. We have calculated the correlation coefficients using the "corrcoef" function in Matlab. For measurements representing cell populations, we imaged enough cells to represent the characteristics of the whole cell population. We selected arbitrary areas to image and analyze to ensure unbiased selection. In cases where the field of view (FOV) contained multiple cells, we analyzed and reported every cell in the FOV to further

minimize selection bias. Also, estimation of OOP of individual cells by visually inspecting their fluorescent image is extremely inaccurate and unpredictable, so it is unlikely that any selection bias would be incorporated while capturing fluorescent images. OOP data from cells were only rejected if the cells showed clear signs of being unhealthy and in rare cases where the analysis showed a completely erratic extraction pattern. In case of live-cell trajectories, we started imaging cells when they showed disorganized stress fiber patterns and tracked them for 24–48 h after EMT induction without any fore-knowledge of how the OOP would change with time. To minimize any selection bias, we reported data for all cells that we were able to successfully track. Cells that became unhealthy or died within the time window of imaging were rejected. For the FAK images, we randomly chose 12 cells (from the entire dataset) to demonstrate the reorganization of FAK spots. We chose five cells each for the supplementary images (Supplementary Figs. 6 and 7) for esthetic purposes.

**Reporting summary**. Further information on research design is available in the Nature Research Reporting Summary linked to this article.

## Data availability

The data that support the findings of this study are available from the corresponding author upon request. All individual data points used to generate the figures in the main manuscript have been added as Supplementary Data 1, each sheet in the excel file corresponds to individual figures (or specific panels) from the manuscript. The unprocessed and uncropped western blot images have been added as Supplementary Fig. 2 in the "Supplementary Figures" file.

## Code availability

The code used for morphological components analysis is available at https://fadili.users.greyc.fr/demos/WaveRestore/downloads/mcalab/Home.html. The codes for fiber extraction and OOP calculation, image cropping, data visualization, and tabulation are available at https://github.com/arkaprabha/Statistical-Parametrization-of-Cell-Cytoskeleton-SPOCC (https://doi.org/10.5281/zenodo.6374491).

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

## Acknowledgements

The authors thank Dr. Michael Lake for help with AFM experiments and Ms. Maya Segal for her help in the organization and writing of this manuscript. We acknowledge the support of the Nano and Pico Characterization Lab at California NanoSystems Institute for the AFM experiments. The research was supported by the STROBE National Science Foundation Science and Technology Center, Grant No. DMR-1548924 and Willard Chair funds.

## Author contributions

A.B. and S.W. designed the research in consultation with M.P. and S.D. A.B. carried out the experiments, data analysis, and interpretation. M.P. did the western blot experiment and corresponding analysis. S.D. provided the cell lines and drugs used. M.A. and H.S. developed the basic stress fiber extraction algorithm. A.G. developed the code for OOP calculations. A.B. and S.W. wrote the manuscript.

## Competing interests

The authors declare no competing interests.
