## [Peer Review File · Communications Biology]

Reviewers' comments:

Reviewer #1 (Remarks to the Author):

Review of Basu et al. „Statistical parameterization of cell cytoskeleton (SPOCC) reveals novel lung cancer cytoskeletal phenotype with partial EMT signature“

The study aims to identify and characterize the states in Epithelial-mesenchymal Transition (EMT) based on cytoskeletal rearrangement. The study focuses on actin filaments, extracted with a framework already proposed by some of the co-authors (ref. 30). The rearrangements and alignments of actin filaments is quantified by a here-proposed Orientational Order Parameter (OOP). The effectiveness of the OOP to distinguish effects of drugs that inhibit different processes governing EMT is also illustrated. While the work is sound, the reviewer has some comments that need to be addressed before considering the study.

Methods

1. The approach for image analysis can be described in one additional paragraph, so that an interested reader does not have to go to ref. 30 to understand the details.
2. In the original study, ref. 30, fibers were stitched together if they overlapped; here it seems that a different rule is used. Please, specify.
3. The precise mathematical definition of OOP, the main contribution of the work, is nowhere provided. This needs to be included and studies where a similar technique has been used for orientation of cytoskeletal elements should be included.
4. The claim that the lengths of fibers do not differ need to supported by respective quantification and test.
5. The code should be provided on GitHub to ensure reproducibility

Abstract

1. The sentence ... “to characterize and quantify drug responses” is not clear in the context; drug responses with respect to cytoskeleton?
2. “intermediate phenotype” is not clear from the context, since intermediate EMT is used before.
3. Stiffness of ... phenotype makes no sense; the authors probably mean stiffness of cells?!

Introduction

1. What does a partial EMT phenotype on line 70 refer to?

Results

1. While the meaning of OOP is intuitively described, the precise quantification is missing. Was the coefficient of variation used or any other measure of dispersion?
2. The claim that OOP is more expressive than other coarse measures needs to be better substantiated. The only comparison involves the aspect ratio and only on a single cell. The authors should clearly illustrate the added value of OOP by deeper comparative analyses.
3. OOP does not seem to be gradually increasing, contrary to the claim on line 153, especially when looking at individual cells (Cell 1, Supp. Fig. 4). Some clarifications are needed at this point.
4. What is also not clear is whether the statistical comparisons shown in Fig. 4f pool the OOP values over the entire time or this is conducted at the last time point. Statistical comparisons at individual time points should be provided for the comparisons of drug effects in Fig. 4g (whenever the time points can be aligned). This analysis could provide data on the duration to the intermediate state.
5. In the discussion, the authors may suggest some ideas for a biophysical model of how the transition takes place to ensure the alignment.

Reviewer #2 (Remarks to the Author):

The authors introduce a computational method for automated quantification of actin stress fiber alignment in fluorescence images of cultured cells. The use this method to examine lung cancer cell lines induced experimentally to undergo epithelial-to-mesenchyme transition. They show their method can detect changes in stress fiber organization during EMT and with experimental manipulation of signaling pathways regulating actin dynamics. They conclude that they have identified a new intermediate EMT phenotype associated with decreased invasiveness. They

propose that their method can provide a high-throughput screening method for agents that might impair metastasis by disrupting the actin reorganization required for EMT.

The paper is written clearly and the quantitative method appears to be a sensitive measure of stress fiber organization. However, I have significant concerns about significance and rigor. The notion that EMT is a "non-binary process" that proceeds through intermediate states does not seem profound or surprising. The claim that the method is high-throughput and non-destructive is undercut by the fact that most of the experiments are performed using fixation and immunostaining and some of the data are based on analyses of small numbers of cells. The conclusion that the intermediate phenotype has decreased invasiveness is based on an inference because the authors do not examine motility or invasiveness directly. The authors claim that their method is superior to other methods such as motility and invasiveness assays or assays of cell asymmetry, but they have not demonstrated this experimentally. In addition, there is a lack of repeated measures and statistical analysis in multiple experiments. For all these reasons, the authors have not convinced me of the utility of their method as a tool for screening potential therapeutic agents that target metastasis.

Specific concerns

1. Line 128. It is not rigorous to make the argument that the OOPS measurements are a superior alternative to the aspect ratio measurements as a marker of cytoskeletal remodeling based on only two cells. A statistical approach is required. The authors should measure the OOP and aspect ratios of a large number of cells (making both measurements on each cell) and then compare the data statistically with respect to some independent measure of EMT. On line 131 the authors write that "the OOP value calculated using SPOCC is a more relevant cell state marker during the EMT" but they have not shown this.
2. Fig. 1h,i. The authors show two examples of cells with similar actin stress fiber alignment but different aspect ratios. The authors' characterization of these aspect ratios (1.66 and 2.99) as "completely different" is not rigorous. "Completely different" might be aspect ratios of 1.0 versus 2.99, or cells in which the actin filaments are orientated parallel and perpendicular to the long axis. It would be more accurate to say that the extent of elongation of the two cells differs, but both cells are elongated in the orientation of the stress fibers.
3. Fig. 2. The authors state that cells with low OOP values had fewer focal adhesions (FAK spots) but this is not clear from the figure shown and no statistical analysis is provided. To my eye, I cannot reject the possibility that the number of focal adhesions is the same and that all that has changed is their distribution. The authors also state that focal adhesions were associated with both ends of stress fibers in cells with high OOP values, but only with one end in cells with low OOP values. Again, there is no quantification or statistical analysis to support this assertion. As above, the lack of quantification and statistical analysis is a weakness.
4. The data in Fig. 4 e,f and Fig. 5 are not subjected to statistical analysis. In fact, there is a general lack of statistical analysis throughout, which is not rigorous.
5. Line 148. It is not clear to me what the authors mean when they say that the two morphological appearances observed in their cultures during EMT could represent distinct cell populations that arise in parallel. Are they suggesting that the cells with low and high OOP values arise independently? If so, what would be the fate of the individual cells with low OOP values as the population undergoes EMT? The notion seems implausible and possibly a bit of a straw man. It is well established that cells reorganize their shape, motility and actin organization dynamically during EMT. More explanation is required.
6. Fig 3 shows the OOP value transitions for a single cell tracked live, but one cell is insufficient to make this point. This is not rigorous. Measurements of additional cells are shown in supplemental Fig 4 but it is not clear why those cells are not included in Fig 3, as it is important to show that the observation is robust and representative of the behavior of the population. Moreover, if this method is truly high-throughput it should be easy to analyze dozens of cells.

7. Suppl. Fig 2. The H260 cells appear to undergo the actin reorganization without losing their cell attachments. This suggests that the actin reorganization can be uncoupled from the classic EMT transition. The authors should comment on this.

Additional suggestion

8. The schematic in Fig 1a would be more effective if the authors removed the purple-to-cyan color transition of the cells, which is unnecessary, and replaced it with a schematic representation of actin reorganization – from cortical to stress fibers. This would emphasize the central focus of this paper, which is the actin reorganization during EMT.

We thank the reviewers for their valuable comments. We do believe that including their suggestions have improved this manuscript thoroughly.

All line numbers listed with the replies correspond to when one is actively viewing the “track changes” function. The line numbers might vary if the changes are hidden.

All figure that have been modified or added are attached at the end of this document (with their respective legends and numbers as they appear on the manuscript)

Reviewers' comments:

Reviewer #1 (Remarks to the Author):

Review of Basu et al. „Statistical parameterization of cell cytoskeleton (SPOCC) reveals novel lung cancer cytoskeletal phenotype with partial EMT signature“

The study aims to identify and characterize the states in Epithelial-mesenchymal Transition (EMT) based on cytoskeletal rearrangement. The study focuses on actin filaments, extracted with a framework already proposed by some of the co-authors (ref. 30). The rearrangements and alignments of actin filaments is quantified by a here-proposed Orientational Order Parameter (OOP). The effectiveness of the OOP to distinguish effects of drugs that inhibit different processes governing EMT is also illustrated. While the work is sound, the reviewer has some comments that need to be addressed before considering the study.

Methods

1. The approach for image analysis can be described in one additional paragraph, so that an interested reader does not have to go to ref. 30 to understand the details.

We have expanded on the image analysis section and added a brief description of the image analysis pipeline. (Lines 416-434)

2. In the original study, ref. 30, fibers were stitched together if they overlapped; here it seems that a different rule is used. Please, specify.

Only fibers that overlap and have same angles are stitched together. We have clarified this point in the same paragraph. (Lines 432-433)

3. The precise mathematical definition of OOP, the main contribution of the work, is nowhere provided. This needs to be included and studies where a similar technique has been used for orientation of cytoskeletal elements should be included.

We agree that adding the mathematical definition of OOP will enhance the reader's understanding of the work. We have made a separate OOP section in the methods part, expanded on our explanation of OOP and included relevant equations. (Lines 435-448)

This particular method of quantification of the cytoskeleton (and in extension, a biological process) is the primary novelty of our work. But we have cited a few studies that have used similar approaches to ours in the Discussion section of the paper. (Lines 291-300)

4. The claim that the lengths of fibers do not differ need to supported by respective quantification and test.

We acknowledge the requirement for demonstrating the fiber length variation in cells. We have added a supplementary figure (Supplementary Fig. 11) in our manuscript to address this concern. (Line 447)

In particular, we show that the mean fiber lengths of 304 A549 cells are of the same order of magnitude with the standard deviation of their distribution as $2.3 \mu\text{m}$ (mean of distribution is $11.2 \mu\text{m}$). For cells which are usually $50\text{-}100 \mu\text{m}$ in size, we believe differences in the order of $2.3 \mu\text{m}$ can be negligible.

5. The code should be provided on GitHub to ensure reproducibility.

We thank the reviewer for this suggestion and we have put the code on GitHub and made the repository public. The code availability statement has been updated in the manuscript. (Lines 478-481)

<https://github.com/arkaprabha/Statistical-Parametrization-of-Cell-Cytoskeleton-SPOCC>

Abstract

1. The sentence ... “to characterize and quantify drug responses” is not clear in the context; drug responses with respect to cytoskeleton?

We have clarified the language to specify that we are referring to cytoskeletal response to the drugs. (Line 27)

2. “intermediate phenotype” is not clear from the context, since intermediate EMT is used before.

We have replaced “intermediate phenotype” with “intermediate EMT phenotype”. We hope, this should communicate the meaning better. (Line 29)

3. Stiffness of ... phenotype makes no sense; the authors probably mean stiffness of cells?!

Here we do refer to cells belonging to a certain phenotype. Phenotype and cell are often used interchangeably in biological contexts.

Introduction

1. What does a partial EMT phenotype on line 70 refer to?

Here we refer to intermediate states that have been reported previously in EMT. These states are neither completely mesenchymal, nor epithelial and as such have been referred to as partial EMT phenotypes or partial EMT states. We have adjusted the language and included this explanation in the manuscript. (Lines 71-74).

Results

1. While the meaning of OOP is intuitively described, the precise quantification is missing. Was the coefficient of variation used or any other measure of dispersion?

We have added the precise definition of OOP in the methods section. We used the tensor method to calculate the OOP. (Lines 435-448)

2. The claim that OOP is more expressive than other coarse measures needs to be better substantiated. The only comparison involves the aspect ratio and only on a single cell. The authors should clearly illustrate the added value of OOP by deeper comparative analyses.

We appreciate suggestion of further quantitative comparison between aspect ratio/fluorescent intensity and OOP. We have compared and analyzed multiple cells and added two new supplementary figure to our manuscript (Supplementary Figs. 4 and 5).

Comparing intensities of different cells can cause in erroneous results as the intensity of each cell depends on its size and labelling density. So we have looked at the intensity trajectories of 9 live cells (Supplementary Fig. 4) (Line 135) which show that their intensities can increase, decrease or remain similar rather randomly. We normalized the intensity series of individual cells with respect to the total intensity of the first image of that cell. We have also compared the OOP and intensity trajectories of 4 of

those cells to support our claim that though OOP increases overall for those cells, the intensity does not follow any given pattern. (Lines 134-138)

We have compared the aspect ratio and OOP of 168 cells with OOP greater than 0.5, which we believe is a statistically significant population. We calculated the correlation coefficient between aspect ratio and OOP of cells, which is 0.23 demonstrating a low degree of correlation. We have also demonstrated that upon considering 66 cells with OOP higher than 0.75, the correlation coefficient drops to 0.05. This demonstrates the aspect ratio becomes more and more unreliable as we consider cells with higher OOP values. These plots are attached with the manuscript as a new supplementary figure (Supplementary Fig. 5). We have also calculated the correlation coefficient of the mean aspect ratio and mean OOP with respect to time-points between 0 and 24 hours. The correlation coefficient between mean OOP and time is 0.95 whereas the correlation coefficient between mean aspect ratio and time is 0.45. We hope that this conclusively demonstrates that OOP is a significantly better indicator of EMT compared to aspect ratio. (Line 139)

3. OOP does not seem to be gradually increasing, contrary to the claim on line 153, especially when looking at individual cells (Cell 1, Supp. Fig. 4). Some clarifications are needed at this point.

We thank the reviewer for pointing out the discrepancy. We have replaced the Supplementary Figure 4 with a new figure (now Supplementary Fig. 8) showing the gradual increase of OOP of 9 live cells. Though individual cells behave differently from one another, it is clear that over all their OOPs increase with time. We have also added a video of a cell undergoing EMT with the corresponding increase in OOP (Supplementary Video 1).

The previous error that the reviewer pointed out resulted from a manual error in plotting the data.

4. What is also not clear is whether the statistical comparisons shown in Fig. 4f pool the OOP values over the entire time or this is conducted at the last time point. Statistical comparisons at individual time points should be provided for the comparisons of drug effects in Fig. 4g (whenever the time points can be aligned). This analysis could provide data on the duration to the intermediate state.

In Fig. 4f, all cells correspond to end point analysis. We have changed the language in the manuscript to better explain our experiments. (Lines 213-223)

We thank the reviewer for suggesting the statistical analysis at individual time-points. They were done for drug effects in Fig. 4g and has been added to the manuscript as a new supplementary figure (Supplementary Fig. 10). (Line 218) Our analysis demonstrates that the difference in OOP between with and without drug cells essentially become significant after 16 hours.

5. In the discussion, the authors may suggest some ideas for a biophysical model of how the transition takes place to ensure the alignment.

The reviewer's suggestion of building a possible biophysical model is extremely relevant, albeit challenging. We have included in our discussion section the reported mechanisms of formation of ventral stress fibers from dorsal fibers and transverse arcs. (Lines 306-310) To build a hypothetical model of fiber alignment, we have drawn parallels between previous studies on cell-stretching with the loss of polarization in EMT. Though this is a plausible hypothesis, we currently do not possess the means of validating or rejecting it. (Lines 324-331)

Reviewer #2 (Remarks to the Author):

The authors introduce a computational method for automated quantification of actin stress fiber alignment in fluorescence images of cultured cells. They use this method to examine lung cancer cell lines induced experimentally to undergo epithelial-to-mesenchyme transition. They show their method can detect changes in stress fiber organization during EMT and with experimental manipulation of signaling

pathways regulating actin dynamics. They conclude that they have identified a new intermediate EMT phenotype associated with decreased invasiveness. They propose that their method can provide a high-throughput screening method for agents that might impair metastasis by disrupting the actin reorganization required for EMT.

The paper is written clearly and the quantitative method appears to be a sensitive measure of stress fiber organization. However, I have significant concerns about significance and rigor. The notion that EMT is a “non-binary process” that proceeds through intermediate states does not seem profound or surprising.

While we appreciate the reviewer’s comment regarding previous reports of intermediate EMT states, we have not claimed that our work is the first report of intermediate EMT states. We have just demonstrated that there is an intermediate state with a unique cytoskeletal signature and that we have developed a tool to quantify, identify and track biological processes. We have claimed that end point studies of stress-fiber formation in EMT is likely to miss any such intermediate cytoskeletal phenotypes. This nest-like phenotype might correspond to an intermediate state previously reported based on RNA or protein expression studies.

The claim that the method is high-throughput and non-destructive is undercut by the fact that most of the experiments are performed using fixation and immunostaining and some of the data are based on analyses of small numbers of cells.

We acknowledge that we have used fixed cells. But we have demonstrated the non-destructive nature of the technique by tracking single live cells undergoing EMT. We accept that our live cell data is limited due to experimental constraints, but the method itself can be used on a large number of live cells in future studies.

The conclusion that the intermediate phenotype has decreased invasiveness is based on an inference because the authors do not examine motility or invasiveness directly.

The reviewer has correctly pointed out that we have drawn an inference on the motility of these cells rather than actually measuring it. But, unfortunately, the available motility (or scratch wound) assays usually average cell movement for 12-24 hours. As this particular phenotypic transition takes place on a time-scale that is comparable to the assay times, it will not be able to measure the motility of the intermediate phenotype (it will basically give us the average motility of the entire phenotypic transition rather than any particular phenotype).

The authors claim that their method is superior to other methods such as motility and invasiveness assays or assays of cell asymmetry, but they have not demonstrated this experimentally. In addition, there is a lack of repeated measures and statistical analysis in multiple experiments. For all these reasons, the authors have not convinced me of the utility of their method as a tool for screening potential therapeutic agents that target metastasis.

We do not propose our technique as an alternative to motility or invasiveness assays. We believe it to be complimentary. While we agree that such assays can provide data on physical properties of cells that our method cannot. On the other hand, SPOCC can provide sub-cellular information that cannot be obtained by motility or invasion assays. Also, these assays take a long time and thus is rendered less useful for dynamic biological processes. In future studies SPOCC can be used in combination with motility assays to develop a correlation between the actin architecture and cellular motility.

Specific concerns

1. Line 128. It is not rigorous to make the argument that the OOPS measurements are a superior alternative to the aspect ratio measurements as a marker of cytoskeletal remodeling based on only two cells. A statistical approach is required. The authors should measure the OOP and aspect ratios of a large number of cells (making both measurements on each cell) and then compare the data statistically

with respect to some independent measure of EMT. On line 131 the authors write that “the OOP value calculated using SPOCC is a more relevant cell state marker during the EMT” but they have not shown this.

The reviewer has made a very appropriate comment regarding the statistical quantification and comparison between aspect ratio and EMT. We have compared and analyzed multiple cells and added a new supplementary figure to our manuscript (Supplementary Fig. 5). (Line 139)

We have compared the aspect ratio and OOP of 168 cells with OOP greater than 0.5, which we believe is a statistically significant population. We calculated the correlation coefficient between aspect ratio and OOP of cells, which is 0.23 demonstrating a low degree of correlation. We have also demonstrated that upon considering 66 cells with OOP higher than 0.75, the correlation coefficient drops to 0.05. This demonstrates the aspect ratio becomes more and more unreliable as we consider cells with higher OOP values. These plots are attached with the manuscript as a new supplementary figure (Supplementary Fig. 5). We have also calculated the correlation coefficient of the mean aspect ratio and mean OOP with respect to time-points between 0 and 24 hours. The correlation coefficient between mean OOP and time is 0.95 whereas the correlation coefficient between mean aspect ratio and time is 0.45. We hope that this conclusively demonstrates that OOP is a significantly better indicator of EMT compared to aspect ratio.

2. Fig. 1h,i. The authors show two examples of cells with similar actin stress fiber alignment but different aspect ratios. The authors’ characterization of these aspect ratios (1.66 and 2.99) as “completely different” is not rigorous. “Completely different” might be aspect ratios of 1.0 versus 2.99, or cells in which the actin filaments are orientated parallel and perpendicular to the long axis. It would be more accurate to say that the extent of elongation of the two cells differs, but both cells are elongated in the orientation of the stress fibers.

In trying to address the comment regarding characterization of aspect ratios, we have corrected Fig. 1h,i to include two cells both of which have OOP values of 0.84 but their aspect ratios are 1.07 and 3.00. We hope that this, coupled with the added Supplementary Fig. 5, will support our claim that “cells with well-aligned fibers can have completely different aspect ratios”.

3. Fig. 2. The authors state that cells with low OOP values had fewer focal adhesions (FAK spots) but this is not clear from the figure shown and no statistical analysis is provided. To my eye, I cannot reject the possibility that the number of focal adhesions is the same and that all that has changed is their distribution. The authors also state that focal adhesions were associated with both ends of stress fibers in cells with high OOP values, but only with one end in cells with low OOP values. Again, there is no quantification or statistical analysis to support this assertion. As above, the lack of quantification and statistical analysis is a weakness.

We agree that we have been unable to quantify the amount of FAK spots and their co-localization with the stress fibers. We have changed the language in the manuscript so as to make no comment about the number of FAK spots in the cells. (Line 164) But we have added two new supplementary figures (Supplementary Figs. 6 and 7) (Line 163) where we show more cells with low and high OOP values. In the low OOP cases, the FAK spots are near the edge of the cells whereas in the high OOP cells they are all over the cells. In case of FAK capping of stress fibers, we rely on visual inspection of the cells to identify the FAK caps from the overlay images. We believe the added supplementary figures support the claim we made in the manuscript.

4. The data in Fig. 4 e,f and Fig. 5 are not subjected to statistical analysis. In fact, there is a general lack of statistical analysis throughout, which is not rigorous.

We thank the reviewer for suggesting statistical analysis in these figures. We have corrected Fig. 4 e,f and Fig. 5 to include the corresponding p-values. Our analysis demonstrates that all the differences we see in these cases are significant.

The details of the statistical analysis have been added in the methods section named “Statistical analyses”. (Lines 455-459)

5. Line 148. It is not clear to me what the authors mean when they say that the two morphological appearances observed in their cultures during EMT could represent distinct cell populations that arise in parallel. Are they suggesting that the cells with low and high OOP values arise independently? If so, what would be the fate of the individual cells with low OOP values as the population undergoes EMT? The notion seems implausible and possibly a bit of a straw man. It is well established that cells reorganize their shape, motility and actin organization dynamically during EMT. More explanation is required.

The reviewer here has brought forward a very interesting point regarding the proposal of possible models. Based on the data we presented in Figs. 1 and 2, we suggested that two possible models can give rise to such data. Because there are multiple steps in the EMT cascade and more than one of the genes involved can affect the actin architecture with different levels of cross-talk, it is not impossible that one of the genes that is activated earlier results in the low OOP cells, whereas another gene activated later results in high OOP cells. We do not claim equal biological validity of the two models. The following experiments were designed and carried out to validate one model over the other. We indeed confirmed the dynamic reorganization of stress fibers from nest-like to semi-parallel arrangement in the live cell experiments described in Fig. 3. As we rejected the hypothesis where low OOP cells retained their low OOP values throughout EMT, any comment on their fate at the end of EMT would be incorrect.

In summary, we posed a question about the two proposed models based on our initial data and then in the following section, we resolved the question. While we agree that low OOP cells retaining low OOP throughout EMT is not very likely based on biological precedents, but it was indeed necessary to rule out alternate possibilities (even unlikely ones) before making conclusions regarding the model.

We have corrected the manuscript and added a brief explanation of the two models in the manuscript. (Lines 181-187)

6. Fig 3 shows the OOP value transitions for a single cell tracked live, but one cell is insufficient to make this point. This is not rigorous. Measurements of additional cells are shown in supplemental Fig 4 but it is not clear why those cells are not included in Fig 3, as it is important to show that the observation is robust and representative of the behavior of the population. Moreover, if this method is truly high-throughput it should be easy to analyze dozens of cells.

We appreciate the suggestion of including multiple single cell trajectories in Fig. 3. In the original manuscript, we had only one single cell trajectory in this figure because we were also showing the fluorescent actin image of that cell in the same figure. But following the reviewer's suggestion, we have corrected Fig. 3 and included all the single cell trajectories in this figure. To identify the cell whose fluorescent image is shown, that trajectory is red whereas the others are grey. We have also corrected Supplementary Fig. 4 (now Supplementary Fig. 8) to show 9 individual single cell trajectories. We have also added a video of a cell undergoing EMT with the corresponding increase in OOP (Supplementary Video 1).

7. Suppl. Fig 2. The H260 cells appear to undergo the actin reorganization without losing their cell attachments. This suggests that the actin reorganization can be uncoupled from the classic EMT transition. The authors should comment on this.

The reviewer has rightly pointed out that H460 cells do not lose attachment. The loss of cell-cell adhesion during EMT results from loss of E-cadherin. It is reported that different cells lose E-cadherin to different extents. Also, there are reports of collective migration of cancer cells where they migrate as a group rather than single cells (DOI: 10.1115/1.4035121). We believe these reasons combined result in formation of groups in H460 cells. Also, we agree that this is fascinating and can answer some deep biological questions, but unfortunately trying to conclusively answer this question will require a substantial amount of work that is beyond of the scope of this paper.

Additional suggestion

8. The schematic in Fig 1a would be more effective if the authors removed the purple-to-cyan color transition of the cells, which is unnecessary, and replaced it with a schematic representation of actin reorganization – from cortical to stress fibers. This would emphasize the central focus of this paper, which is the actin reorganization during EMT.

Fig. 1a is meant to introduce the general aspects of EMT and not specifically the cytoskeletal rearrangement. Following the reviewer's suggestions, we have added an extra figure (Fig. 6) in the manuscript, which demonstrates the actin reorganization as well as summarizes our findings. (Line 265)

Modified and Added Figures:

Fig. 1 | Identification and quantification of cytoskeletal phenotypes. **a**, Cartoon image of cells undergoing EMT with formation of stress fibers and up/downregulation of proteins. **b, c, d**, Fluorescent images of A549 cells stained with phalloidin after 0, 14 and 48 hours of TGF β 1 induced EMT respectively. **e, f**, Angular distribution of stress fibers and corresponding Orientational Order Parameter (OOP) values for cells shown in **c** and **d** respectively. **g**, Plot of OOP values for a cell population against time of TGF β 1 treatment. Mean values are reported and error bars correspond to standard error values for every time point. n=12 (for 0hours), n=15 (for 1hour), n=13 (for 2hours), n=20 (for 4hours), n=19 (for 6hours), n=20 (for 8hours), n=22 (for 10hours), n=20 (for 12hours), n=22 (for 14hours), n=22 (for 16hours), n=20 (for 18hours), n=19 (for 20hours), n=23 (for 22hours), n=20 (for 24hours) and n=19 (for 48hours). **h, i**, Fluorescent images of two cells stained with phalloidin with highly aligned stress fibers (similar OOP values) but drastically difference aspect ratios. Scale bar: 16 μ m.

Fig. 3 | Tracking phenotypic transition in single cell stained with SiR-actin. **a, b**, A single A549 cell stained with SiR-actin after 12 hours and 48 hours of TGF β 1 addition respectively. **c, d**, Extracted stress fiber image from **a** and **b** respectively. **e, f**, Angular distribution and OOP values of the cells shown in **a** and **b** respectively. **g**, Plot of OOP values of a live cell (shown in **a-f**) against time of TGF β 1 treatment (red). Multiple live cell OOP trajectories (grey) against time of TGF β 1 treatment. Scale bar: 16 μ m.

Fig. 4 | Quantification of drug response of EMT over 48 hours. a-d, A549 cells stained with phalloidin after 48 hours of EMT induction in the presence of no drug (a), Rhosin (b), JNK 1/2 Inhibitor (c) and Tankyrase Inhibitor (d). e, Boxplot comparison of the number of stress fibers extracted from control (no drug) cells vs Rhosin treated cells. Red lines inside the boxes correspond to the median values, the bottom and top edges of the boxes correspond to the 25th and 75th percentiles respectively, the whiskers are extended to the most extreme data that is not considered an outlier in each direction. f, Plot showing the OOP values of control cells (green) and cells treated with JNK 1/2 inhibitor (blue) and Tankyrase inhibitor. Mean mean of the distributions are shown as larger dots of the same color and the error bars correspond to standard error values. g, Plots of mean OOP values against time of a cell population undergoing EMT with (red) and without (blue) the presence of Tankyrase Inhibitor. Error bars correspond to standard errors at each time point. For untreated cells, the sample size at each time point is same as reported in Fig. 1g. For Tankyrase inhibited cells: n=9 (for 0hours), n=12 (for 2hours), n=11 (for 4hours), n=9 (for 6hours), n=11 (for 8hours), n=11 (for 10hours) n=11 (for 12hours), n=11 (for 14hours), n=10 (for 16hours), n=12 (for 18hours), n=10 (for 24hours), n=11 (for 36hours) and n=9 (for 48hours). h, Plots of OOP values against time of a single cell undergoing EMT with (red) and without (blue) the presence of Tankyrase Inhibitor. Scale bar 16 μ m. $p_1 = 1.85 \times 10^{-6}$, $p_2 = 1.28 \times 10^{-10}$, $p_3 = 6.24 \times 10^{-6}$.

Fig. 5 | Measurement of elastic properties of cells. Young's Modulus values at 0 (blue) ($n = 15$), 14 (brown) ($n=13$) and 48 (green) ($n=14$) hours after EMT induction using $TGF\beta 1$ showing their mean and standard errors along with a plot of the mean values (red). Young's modulus values were extracted from AFM force curve measurements. Plot of mean OOP values for cell populations against hours after EMT (red) showing mean values and standard errors at 0, 14 and 48 hour time points. Sample sizes for OOP values are same as reported in Fig. 1g. $p_1 = 0.0012$, $p_2 = 0.0048$, $p_3 = 3.6 \times 10^{-9}$.

Fig. 6 | Schematic of the cytoskeletal reorganization, relevant genetic pathways and increase of OOP during Epithelial-Mesenchymal Transition of A549 cells.

Supplementary Fig. 4 | Variation of total intensity of live cells undergoing EMT. a, Plot of normalized total intensity of fluorescent actin images vs. hours of TGF β 1 treatment demonstrating random variation in intensities. The intensity series for each cell was normalized with respect to the intensity of the first image of that cell. **b-e,** Plots of OOP and total intensities of Cell 1, Cell 2, Cell 3 and

Cell 4 respectively showing four cells with overall increasing OOPs can have different intensity trajectories.

Supplementary Fig. 5 | Distribution of Aspect Ratio of cells with respect to their OOPs. **a**, Plot of aspect ratio of cells ($n = 168$) vs. their corresponding OOPs ($OOP > 0.5$) demonstrating poor correlation between aspect ratio and OOP (Correlation Coefficient = 0.23). **b**, Plot of aspect ratio of cells ($n = 66$) vs. their corresponding OOPs ($OOP > 0.75$) demonstrating correlation between aspect ratio and OOP getting worse for cells with higher OOP (Correlation Coefficient = 0.05). **c**, Plot of mean aspect ratio (Correlation Coefficient: 0.45) and OOP (Correlation Coefficient: 0.91) of cells ($n = 297$) against hours of TGF β 1 treatment.

Supplementary Fig. 6 | Focal Adhesion Kinase (FAK) pattern of multiple low OOP cells. Fluorescent image of cell with disoriented actin stress fibers (magenta, left panels), fluorescent image of FAK (green, middle panels) of the same cell shown in left panels FAK spots near the cell-edge. Overlay image (right panels) of actin (magenta) and FAK (green) of the cells showing stress fibers with zero or one FAK capping. Scale bar: 16 μ m.

Supplementary Fig. 7 | Focal Adhesion Kinase (FAK) pattern of multiple high OOP cells.

Fluorescent image of cell with disoriented actin stress fibers (magenta, left panels), fluorescent image of FAK (green, middle panels) of the same cell shown in left panels FAK spots throughout the cell. Overlay image (right panels) of actin (magenta) and FAK (green) of the cells showing stress fibers with one or two FAK capping. Scale bar: 16 μ m.

Supplementary Fig. 8 | Plot of multiple single cell OOP trajectories with time of EMT induction.

Supplementary Fig. 9 | Comparison of length of actn fibers extracted from fluorescent images. Every length is normalized with respect to the length of the longest extracted fiber. Control: n=12. Rhosin: n=18. $p = 7.97 \times 10^{-7}$.

Supplementary Fig. 10 | Statistical comparison of cell populations with and without Tankyrase treatment showing significant difference in OOP from 16-48 hours. We have conducted the T-test on the cell populations with and without Tankyrase at the same time points. The h-value (left hand Y-axis) or the binary rejection of null hypothesis (the two data sets belong to independent normal distributions with the same mean) shows that the hypothesis is rejected for time-points 16 hours and later, i.e., after 16 hours, drug treatment creates a significant difference on the OOP values. The p-value (right hand Y-axis) clearly demonstrates that the p-value is larger than the cut-off 0.05 for 0-14 hour time-points indicating that the validity of the null hypothesis for those time-points can occur due to randomness and as such is not significant. p-values on or after 16 hours are all below 0.05 indicating the rejection of the null hypothesis is significant.

Supplementary Fig. 11 | Variation of mean fiber lengths of A549 cells. a, Plot of mean fiber length of 304 cells demonstrating most cells have a mean fiber length between 8-14 μm . **b,** Gaussian fit of mean fiber lengths with a mean 11.2 μm and standard deviation 2.3 μm .

Reviewers' comments:

Reviewer #1 (Remarks to the Author):

The authors have carefully addressed all points raised by the reviewer. In particular, the reviewer acknowledges:

- the newly added methods, to describe the details of the filament extraction as well as definition of the measure used in the presented analyses
- the pointer to GitHub repository, to ensure reproducibility of the findings
- correction of the noted errors in the supplementary figures of the earlier version, and
- addition of supplementary figures to detail the adequateness of the proposed measure.

Reviewer #2 (Remarks to the Author):

The authors' revisions to the manuscript and figures have addressed most of my specific concerns. The addition of statistical tests goes some way to improving the rigor of their analysis, but they should also explain what measures were taken to ensure unbiased selection of cells for their data analysis. This will enhance the reader's confidence in the rigor of their approach and the quality of their data. Examples include Fig. 3 and Supplementary Figs. 4,6,7 and 8. This is of particular concern given the small number of cells that are analyzed/presented in these figures.

Specific comments:

1. Fig. 3 shows live cell data for multiple cell trajectories. This figure is intended to show that the method can be used to track the phenotypic transition in living cells, but it still lacks rigor. The number of cells is not provided but it appears to be 9, based on Supplementary Fig. 8. There is no explanation of how this small number of cells were selected.
2. The authors still claim in the manuscript that their method has "improved throughput" and have not responded to the critique of this statement.
3. The authors note that the time scale of motility assays is such that they cannot correlate their measure of cytoskeletal organization with motility or invasiveness directly. While I appreciate this technical limitation, my concern about the inference still stands. Specifically, in the abstract the authors state "... owing to the increased stiffness (and hence invasiveness) of the intermediate EMT phenotype compared to mesenchymal cells". The authors should revise their language to avoid unintentionally implying that this manuscript presents data on invasiveness.
4. The authors note that they do not propose their technique as an alternative to motility or invasiveness assays. The authors should state this in the Discussion.
5. The new data in Supplementary Fig. 5 is helpful. The data do demonstrate that aspect ratio does not correlate well with OOP, but the authors should explain in the manuscript why they excluded data for OOP values less than 0.5 or 0.75, which seems arbitrary. Why not show the data for all OOP values?
6. Fig. 6 and 7 do show a difference in the distribution of focal adhesions, though the authors still do not provide quantification and it is unclear how the cells in this figure were chosen, which undermines confidence in the data. This could be addressed in the Methods.
7. The authors have added statistical analysis to Figs. 4 and 5 but their method of showing the p values (p_1 , p_2 , p_3) is unfamiliar to me and seems unconventional. The authors should use asterisks according to standard conventions (* for $p < 0.05$, ** for $p < 0.01$, *** for $p < 0.001$, etc).
8. I appreciate the authors' response regarding the fact that the H460 cells undergo cytoskeletal reorganization without losing their cell attachments, and understand why they do not address this experimentally, but suggest that they address this in their Discussion.

We thank the reviewers for taking the time to review our manuscript twice. We believe their insightful suggestions have enriched the quality of our manuscript.

We have highlighted all the changes made to the manuscript in this revision, to distinguish them from the changes made in the earlier revision.

Reviewers' comments:

Reviewer #1 (Remarks to the Author):

The authors have carefully addressed all points raised by the reviewer. In particular, the reviewer acknowledges:

- the newly added methods, to describe the details of the filament extraction as well as definition of the measure used in the presented analyses
- the pointer to GitHub repository, to ensure reproducibility of the findings
- correction of the noted errors in the supplementary figures of the earlier version, and
- addition of supplementary figures to detail the adequateness of the proposed measure.

We are glad to have addressed the reviewer's concern to their satisfaction. We truly believe incorporating the suggestions has improved our manuscripts thoroughly.

Reviewer #2 (Remarks to the Author):

The authors' revisions to the manuscript and figures have addressed most of my specific concerns. The addition of statistical tests goes some way to improving the rigor of their analysis, but they should also explain what measures were taken to ensure unbiased selection of cells for their data analysis. This will enhance the reader's confidence in the rigor of their approach and the quality of their data. Examples include Fig. 3 and Supplementary Figs. 4,6,7 and 8. This is of particular concern given the small number of cells that are analyzed/presented in these figures.

We thank the reviewer for mentioning the point regarding unbiased selection of cells. We have briefly addressed this in the Sample Size and Replication sections of the Nature Research Reporting Summary. We imaged enough cells to represent the overall characteristics of the cell population to avoid any manual bias and as such our sample size varied between experiments. Our analysis pipeline, as of yet, is not fully automated and we cannot analyze all cells in a field of view (FOV) in one batch. We do have to select cells for analysis, and this process could be biased (we plan to automate the pipeline to enhance objective analysis, but this is a big, time consuming task and is outside the scope of the current work). However, for any image where there are more than one cell in the FOV, we imaged and analyzed every single cell that is completely within the FOV. This should also minimize selection bias as we do not expect cells with similar phenotypes to be spatially closer to one another. For example, there are three cells that are completely inside the field of view in Fig. 1h (taken after 24hrs of EMT induction). The three cells have OOP values of 0.58, 0.84 and 0.62. Though two of the three cells had OOP values lower than expected, all three were analyzed and reported. Moreover, it is rather difficult to estimate the OOP value of a cell by looking at the fluorescent image (except in the extreme cases). Therefore, there is a very low chance of introducing selection bias while selecting arbitrary regions for imaging and analysis. Also, for population measurements, multiple experiments were conducted and data from all such experiments were included. For example, in Fig. 1g, 286 cells were reported, which contains data taken over three different experiments and we believe that to be significant enough to represent the behavior of whole cell populations. Another important factor we took into consideration to minimize selection bias was how we rejected certain cells before or after analysis. Cells were only rejected if they were unhealthy or in very rare cases if the analysis showed completely erratic extraction patterns. No measure was taken to include or exclude data based on whether they support the hypothesis. The reviewer's concern of having a small number of live cells (in Fig. 3 and Supplementary Fig. 8) has been addressed in Specific comments #1. In Supplementary Fig. 4, we show the same nine live cells shown in Fig. 3. We only showed the fluorescence intensity trajectory of the live cells because the initial fluorescence intensity can vary from cell to cell (depending on size of cell, staining efficiency etc.) and is subject to bleaching (which

introduces further error) (Lines 135-136), so unless we can compare the intensity of the same cell with time, the data will not be very meaningful. The reviewer's concern regarding Supplementary Fig. 6 and 7 has been addressed in Specific comments #6. We have added a portion on "Selection Criteria" in the "Materials and Methods" section of the manuscript (Lines 478-491).

Specific comments:

1. Fig. 3 shows live cell data for multiple cell trajectories. This figure is intended to show that the method can be used to track the phenotypic transition in living cells, but it still lacks rigor. The number of cells is not provided but it appears to be 9, based on Supplementary Fig. 8. There is no explanation of how this small number of cells were selected.

We appreciate the reviewer's comment regarding the selection criteria. We acknowledge that a higher number of live cell trajectories would have been helpful. But unfortunately, we were limited by our experimental setup where we could only track one cell in each 48-hour experiment, since very high magnification (small field-of-view) is needed for OOP analysis. We also imaged (but not tracked) multiple cells at the beginning and at the end of the experiments to confirm that the increase in OOP of cells was indeed true for the whole population, rather than just the cells we tracked. In terms of unbiased selection of cells, we started imaging cells when they had disorganized stress fiber patterns and reported every cell trajectory that we successfully tracked. We did not have the fore-knowledge of whether the OOP of these individual cells would increase with time or not. We started imaging cells around 10-12 hours after EMT induction when cells already start showing disorganized stress fibers and imaged until 24 to 48 hours. Any cell that became unhealthy, died were rejected. Also, for any case where the cell being tracked could no longer be identified confidently amongst its neighbors (due to shape change and movement), those cells were rejected as well. Therefore, even though we did more live cell experiments, we were only able to track 9 cells until 24-48 hours. We have included this explanation in the "selection criteria" portion of the "Materials and Methods" section (Lines 485-489).

2. The authors still claim in the manuscript that their method has "improved throughput" and have not responded to the critique of this statement.

We agree with the reviewer that we did not provide a clear response to the critique regarding throughput. We do not claim that our technique has higher throughput than all the other techniques listed. For example, our technique is definitely less destructive and less expensive than single cell RNA sequencing, but we do not intend to claim that SPOCC has a higher throughput than high-throughput RNA sequencing. On the other hand, SPOCC has better throughput than traditional techniques such as migration and invasion assays. These assays take time and as such are incapable of identifying fast biological processes. These assays are ultimately dependent on high throughput imaging, but are limited by the fact that the properties being measured here require longer times to manifest. Time-resolution of SPOCC is limited by the fluorescent imaging, which is much faster than the assays. Given the state-of-the-art imaging systems available, SPOCC can definitely achieve higher throughput than migration/invasion assays. We acknowledge that, being limited by our imaging setup, we have not quantitatively demonstrated that SPOCC has a higher throughput than the assays. But SPOCC definitely has the capability of being higher throughput compared to these assays if coupled with a high-throughput imaging setup. We have modified the language in the manuscript to reflect this (Line 77). Further discussions (Lines 312-319) should also clarify which techniques we are comparing SPOCC with to claim improved throughput.

3. The authors note that the time scale of motility assays is such that they cannot correlate their measure of cytoskeletal organization with motility or invasiveness directly. While I appreciate this technical limitation, my concern about the inference still stands. Specifically, in the abstract the authors state "... owing to the increased stiffness (and hence invasiveness) of the intermediate EMT phenotype compared to mesenchymal cells". The authors should revise their language to avoid unintentionally implying that this manuscript presents data on invasiveness.

The reviewer has rightly pointed out the requirement of clearly expressing that the change in invasiveness is based on inference. We have revised the language in the abstract to reflect that our claim on invasiveness is based on inference (Line 29). We have also added the basis and explanation of this inference in the “Discussion” section of the manuscript (Lines 286-288, Lines 312-319). Also, we already explained why we expect the motility to change between the intermediate and mesenchymal phenotype based on the difference of their stress fiber types (Lines 325-328). We hope that these combined should communicate clearly that our claim on modified invasiveness is based on an inference.

4. The authors note that they do not propose their technique as an alternative to motility or invasiveness assays. The authors should state this in the Discussion.

We appreciate the reviewer’s suggestion of clarifying our view on the relation between SPOCC and motility/invasion assays. We have added a paragraph in the “Discussion” section explaining this point as well as potential future applications of combining and correlating measurements from the two techniques (Lines 312-319).

5. The new data in Supplementary Fig. 5 is helpful. The data do demonstrate that aspect ratio does not correlate well with OOP, but the authors should explain in the manuscript why they excluded data for OOP values less than 0.5 or 0.75, which seems arbitrary. Why not show the data for all OOP values?

We thank the reviewer for pointing out that the OOP cut-offs we have used require more explanation. We state in the manuscript that **“cells with well-aligned fibers can have completely different aspect ratios”**. Here we have referred to high OOP cells as cells with well-aligned fibers. It is impossible for a disorganized fiber pattern to exist in a cell with high aspect ratio (there will be fewer fibers in the direction of the minor axis). In other words, low OOP structures are more likely to be supported by a low aspect ratio cell. But the reverse is not true. Cells with aligned stress fibers can have both high and low aspect ratios. That is why we chose the cut-off of 0.5. We have also demonstrated that if we choose cells with even higher degree of alignment of their stress fibers (OOP>0.75), the correlation decreases even further. So, OOP is a better marker for cytoskeletal rearrangement (and hence EMT) than aspect ratio (demonstrated in Supplementary Fig. 5c).

If we plot aspect ratio and OOP for all cells (figure attached below), the correlation coefficient is 0.4, which still shows poor correlation. We are not adding this figure to the manuscript, as our original claim was regarding high OOP cells (cells with high OOP can have both high and low aspect ratio).

In response to the reviewer's comment, we have changed the language in the manuscript to indicate that cells with well-aligned fibers refer to high OOP cells (Line 138).

6. Fig. 6 and 7 do show a difference in the distribution of focal adhesions, though the authors still do not provide quantification and it is unclear how the cells in this figure were chosen, which undermines confidence in the data. This could be addressed in the Methods.

We thank the reviewer for bringing up the selection criteria. Developing and automating an unbiased analysis pipeline to quantify distribution of focal adhesions FAK spots and their co-localization is again a significant endeavor that goes beyond the scope of the current paper. Instead, we added Supplementary Figs. 6 and 7 to demonstrate that the redistribution of FAK spots happens in multiple cells and not just the two shown in Fig. 2. We **randomly** selected five low OOP and five high OOP cells from the datasets and showed them in Supplementary Figs 6 and 7. We have more examples in each category, but putting images of more than five cells in a figure cluttered up the figure. No specific selection criterion was used to select the cells that are shown in Fig. 2 and Supplementary Figs. 6 and 7. We have included this explanation in the "Selection Criteria" of the "Materials and Methods" section (Lines 489-491).

7. The authors have added statistical analysis to Figs. 4 and 5 but their method of showing the p values (p_1 , p_2 , p_3) is unfamiliar to me and seems unconventional. The authors should use asterisks according to standard conventions (* for $p \leq 0.05$, ** for $p < 0.01$, *** for $p < 0.001$, etc).

We appreciate the reviewer mentioning the unconventional reporting of p values. However, in the Statistics section of the Nature Research Reporting Summary, reporting of exact P-values are suggested for null hypothesis testing. We followed this stated guideline.

8. I appreciate the authors' response regarding the fact that the H460 cells undergo cytoskeletal reorganization without losing their cell attachments, and understand why they do not address this experimentally, but suggest that they address this in their Discussion.

In response to the reviewer's appropriate suggestion of including the cell-adhesion point in our manuscript, we have added a paragraph in the "Discussion" section pointing out the phenomenon and providing a plausible explanation based on literature as well as possible future directions (Lines 289-295).

REVIEWERS' COMMENTS:

Reviewer #2 (Remarks to the Author):

The authors have responded admirably to my comments and concerns. I have no remaining concerns.

I do still recommend that the authors reconsider their method of showing the p values in Figs. 4 and 5. The conventional asterisk notation conveys an immediate visual indication of the statistical significance that is widely familiar to readers whereas "p1", p2" etc. mean nothing to me, so I have to refer to the legend to determine the significance. The authors could use the asterisk notation and still include the actual p values in the legends to comply with journal policy. I have seen that approach in many papers. However, this is for the editors and authors to decide.

We thank the reviewers for taking the time to review our manuscript. We believe their insightful suggestions have enriched the quality of our manuscript.

REVIEWERS' COMMENTS:

Reviewer #2 (Remarks to the Author):

The authors have responded admirably to my comments and concerns. I have no remaining concerns.

I do still recommend that the authors reconsider their method of showing the p values in Figs. 4 and 5. The conventional asterisk notation conveys an immediate visual indication of the statistical significance that is widely familiar to readers whereas "p1", p2" etc. mean nothing to me, so I have to refer to the legend to determine the significance. The authors could use the asterisk notation and still include the actual p values in the legends to comply with journal policy. I have seen that approach in many papers. However, this is for the editors and authors to decide.

The reviewer has rightly pointed out the visual identification of the asterisk symbols in reference to statistical significance. Following their suggestion, we have added the asterisk symbols in the figures as well as the exact p-values in the figure legends for both Figs. 4 and 5.